# Sub-nanosecond memristor based on ferroelectric tunnel junction

Chao Ma[1,4], Zhen Luo[1,4], Weichuan Huang[1], Letian Zhao[1], Qiaoling Chen[1], Yue Lin [1], Xiang Liu[1], Zhiwei Chen[1], Chuanchuan Liu[1], Haoyang Sun[1], Xi Jin[1], Yuewei Yin [1]✉ & Xiaoguang Li [1,2,3]✉

Next-generation non-volatile memories with ultrafast speed, low power consumption, and high density are highly desired in the era of big data. Here, we report a high performance memristor based on a Ag/BaTiO$_3$/Nb:SrTiO$_3$ ferroelectric tunnel junction (FTJ) with the fastest operation speed (600 ps) and the highest number of states (32 states or 5 bits) per cell among the reported FTJs. The sub-nanosecond resistive switching maintains up to 358 K, and the write current density is as low as $4 \times 10^3$ A cm$^{-2}$. The functionality of spike-timing-dependent plasticity served as a solid synaptic device is also obtained with ultrafast operation. Furthermore, it is demonstrated that a Nb:SrTiO$_3$ electrode with a higher carrier concentration and a metal electrode with lower work function tend to improve the operation speed. These results may throw light on the way for overcoming the storage performance gap between different levels of the memory hierarchy and developing ultrafast neuromorphic computing systems.

[1] Hefei National Laboratory for Physical Sciences at the Microscale, Department of Physics, and CAS key Laboratory of Strongly-Coupled Quantum Matter Physics, University of Science and Technology of China, Hefei, China. [2] Key Laboratory of Materials Physics, Institute of Solid State Physics, CAS, Hefei, China. [3] Collaborative Innovation Center of Advanced Microstructures, Nanjing, China. [4]These authors contributed equally: Chao Ma, Zhen Luo. ✉email: yyw@ustc.edu.cn; lixg@ustc.edu.cn

In order to efficiently store and process huge amounts of information in the era of big data, researchers keep working on improving memory performances to achieve high-speed operation, low-energy consumption, non-volatility, and high-density integration[1–3]. The next-generation memory should be ultrafast as a static random access memory (SRAM) and possess non-volatility and high-density as a flash memory[4], to overcome the large storage performance gap between different levels of the memory hierarchy. Of all the potential candidates, including phase-change RAM (PCRAM)[5,6], magnetoresistive RAM (MRAM)[7,8], and resistive RAM (ReRAM)[9,10], a recently developed non-volatile resistive-type memory based on ferroelectric tunnel junction (FTJ) shows special advantages[11–14]. Especially, the write current density of an FTJ is as low as $10^3$–$10^4$ A cm$^{-2}$, while the typical values are about $10^6$ A cm$^{-2}$ in the former three types of memories.

Different from the charge-type ferroelectric RAMs (FeRAMs) which suffer from the destructive readout and low storage density[15], the data in FTJs are non-volatilely stored in the ultrathin (typically <5 nm) ferroelectric barrier and can be non-destructively read out via ferroelectric polarization orientation-dependent resistance. In the last decade, high-density multi-state data storage, high-speed and low-energy consumption have been verified in FTJs[14,16,17]. Although the fastest switching time reported at present has reached 6 ns in FTJs[14], the low-temperature operation and small ON/OFF ratio limit its application. Furthermore, to catch up with the microprocessor working in gigahertz lockstep, developing an FTJ with a sub-nanosecond operation speed is required. Based on the working principle of an FTJ[11], its switching time is mainly determined by the ferroelectric domain switching dynamics. In principle, the ferroelectric polarization reversal may theoretically be as fast as around $10^{-13}$ s, approaching optical phonon frequencies[18]. However, for an FTJ with ultrathin ferroelectric film, the situations become very complicated. Many factors neglected in thick ferroelectric films become significantly important in affecting ferroelectric switching especially in sub-nanosecond range, such as electrode effect, interface barriers, depolarization field, domain patterns, size and strain effects, and so on[19]. Thus, it is still unclear whether FTJs could be operated at sub-nanosecond, which is critically important for technical applications.

Recently, a metal/ferroelectric/semiconductor (MFS) type FTJ utilizing n-type Nb-doped SrTiO$_3$ (Nb:SrTiO$_3$) as the semiconductor electrode has drawn considerable attentions, due to its colossal current ON/OFF ratio at room temperature[20]. The significant tunneling electroresistance (TER) effect is raised by an extra Schottky barrier, which can be manipulated by the ferroelectric polarization[21]. Most recently, the optically controlled TER effect and electrically tuned photovoltage were observed in an MFS-FTJ[22], further illustrating its significance. Besides the intrinsic ferroelectricity induced resistive switching, the defect-mediated processes, such as the voltage-driven charge trapping/detrapping and oxygen vacancy migration, can also lead to a colossal TER effect in Nb:SrTiO$_3$-based FTJs[23,24]. However, several reports in Nb:SrTiO$_3$-based FTJs indicated that the time scale of resistive switching induced by ferroelectricity is around 10 ns[14,24,25], while the latter mediated by defects is above 100 ns[24,25]. That is, the investigation of ultrafast resistive switchings is not only essential for the future development of memory technology, but also helpful to deeply understand the resistive switching mechanism in FTJs.

Furthermore, conventional von Neumann computers store and process data separately and have hence imposed fundamental limits on the intelligence, speed, and power efficiency of a computing system[1,2]. To overcome this von Neumann bottleneck, the neuromorphic computing drawing inspiration from the architecture and principle of biological brains is developed to merge the memory and processing functionalities[1,2,26]. The FTJs-based memristors may serve as artificial synapses to construct neuromorphic computational networks[27]. The typical time windows of previous reported artificial synapse are only about $10^2$–$10^6$ ns[27–29], which cannot meet the requirement of higher data processing speed for artificial intelligence. So the realization of the ultrafast artificial synapse emulation is certainly necessary and urgent. In addition, as self-heating effect in integrated circuits is unavoidable and therefore the thermal endurance of the FTJs at the typical operation temperature (358 K) of integrated circuits[30] will be an important issue for practical implement. It is significant to investigate whether the ultrafast switching persists above room temperature or not.

In this work, we constructed the high-performance MFS-type FTJ memristors based on Ag/BaTiO$_3$ (BTO)/Nb:SrTiO$_3$ (NSTO) heterojunctions, and achieved stable non-volatile resistance switchings with the fastest operation speed of 600 ps even up to 358 K. The 32 distinct non-volatile resistive states in one single unit is demonstrated. It shows the fastest operation speed and the highest number of states per cell among the reported FTJs. And the write current density is as low as about $4 \times 10^3$ A cm$^{-2}$, 2–3 orders of magnitude smaller than the other non-volatile memories (PCRAM, MRAM, ReRAM, etc.). The ultrafast spike-timing-dependent plasticity (STDP) is also emulated in the FTJ memristor-based artificial synapse. Furthermore, it is found that the higher Nb concentration in NSTO semiconducting electrode and the lower work function in metal electrode are beneficial to increase the resistive switching speed for the FTJs. Based on the experimental performances of the FTJ memristor, an artificial neural network (ANN) has been simulated by using the stochastic gradient descent (SGD) and back propagation (BP) algorithms, and a high recognition accuracy of >90% on the Modified National Institute of Standard and Technology (MNIST) hand-written digits is obtained through an online supervised learning.

## Results

**Structural and ferroelectric characterizations.** Figure 1a shows an aberration-corrected high-angle annular dark-field scanning transmission electron microscopy (HAADF-STEM) images from the Ag/BTO/NSTO (Nb:0.7 wt%) FTJ. The core-level electron energy-loss spectroscopy (EELS) line scan results for Ba and Ti elements are also displayed. An epitaxial BTO film with a thickness of 6 unit cells (~2.4 nm) and a c/a of about 1.05 is demonstrated. As shown in the inset of Fig. 1a, the Ti ion displacement of about −15 pm along the [001] direction suggests a downward polarization in BTO. Figure 1b shows the abrupt change of the out-of-plane lattice spacing at the BTO/NSTO interface, consistent with the EELS results. Figure 1c, d show the HAADF-STEM images from the Ag/BTO/NSTO FTJs after poling the ferroelectricity upward and downward by −3 and +3 V voltage pulses (pulse duration $t_d$ = 100 ns), respectively. As shown in the insets of Fig. 1c, d, the Ti ion displacements of about 18 and −17 pm along the [001] direction suggest an upward and downward polarized BTO, respectively, which is consistent with the ferroelectric polarization orientations. In addition, the energy-dispersive X-ray spectroscopy (EDS) mapping for Ag element confirms that there is no Ag migration or Ag filament in BTO after poling upward and downward, as shown in Fig. 1e, f.

The piezoresponse force microscopy (PFM)[31] was used to characterize the ferroelectric properties of the ultrathin BTO film. Figure 1g displays the PFM hysteresis loops in amplitude and phase measured from a BTO bare surface. Figure 1h, i show the images of PFM phase and amplitude after the domains patterning

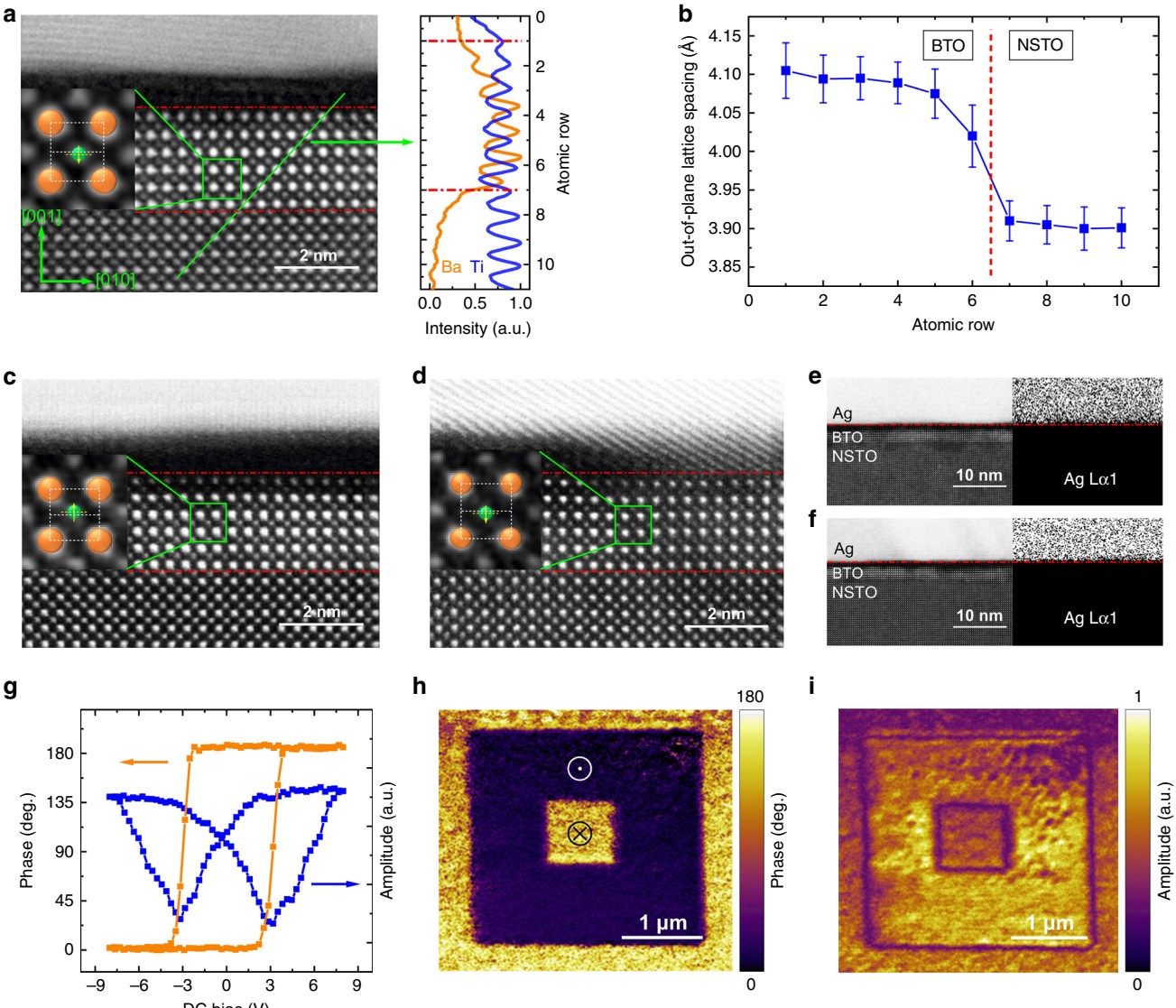

**Fig. 1 Structural and ferroelectric properties. a** Left panel: HAADF-STEM image of the Ag/BTO/NSTO FTJ at the virgin state with the inset showing the Ti ion displacement in BTO. The orange and green spheres denote Ba and Ti ions, respectively. Right panel: Elemental profiles obtained from the EELS of Ba and Ti elements. **b** Mean value of the out-of-plane lattice spacing near the BTO/NSTO interface. The error bars indicate standard deviations. **c, d** HAADF-STEM images of the FTJs after poling the ferroelectricity upward and downward by $-3$ and $+3$ V voltage pulses with $t_d = 100$ ns. Insets show displacements of Ti ions. **e, f** Ag element distributions measured by the EDS mapping for FTJs after poling the ferroelectricity upward and downward. **g** PFM hysteresis loops in phase (orange) and amplitude (blue). **h** PFM phase and **i** PFM amplitude images recorded after writing an area of $3 \times 3\,\mu m^2$ with $-6$ V and then the central $1 \times 1\,\mu m^2$ with $+6$ V by a biased conductive tip.

by $\pm 6$ V. The 180° phase contrast between different record areas reveals the reversible polarizations in the BTO domains. And the virgin state image of PFM for BTO shows a downward polarization, which is consistent with the STEM results in Fig. 1a.

**Sub-nanosecond ultrafast memristor.** The representative $I-V$ curves of the Ag/BTO/NSTO (Nb: 0.7 wt%) FTJs show typical pinched hysteresis loops with a memristive characteristic (Supplementary Fig. 1). To investigate the memristor behaviors at different operating speeds, the resistances (read at 0.1 V) versus voltage pulse ($V_p$) were tested with $V_p$ sweeping in the $0$ V $\rightarrow$ the negative maximum voltage ($V_p^{\max -}$) $\rightarrow$ the positive maximum voltage ($V_p^{\max +}$) $\rightarrow 0$ V sequences (Supplementary Fig. 2a). Here, a $V_p^{\max +}$ was applied before each $R-V_p$ loop measurement and the pulse duration $t_d$ of $V_p$ changes from 600 ps to 100 ns. The $R-V_p$

loops recorded with $t_d = 10$ ns are shown in Fig. 2a, with fixed $V_p^{\max +}$ ($+3.0$ V) and varied $V_p^{\max -}$ ($-2.2$, $-2.6$, $-3.0$, and $-3.4$ V). The lowest resistance state (defined as the ON state) was set by $+3$ V (the ferroelectric polarization points to NSTO) and switched to the highest resistance state (defined as the OFF state) by $-3.4$ V (the ferroelectric polarization points to Ag). The reason is that with the polarization pointing to the semiconducting NSTO, the negative screening charges in the depletion layer tend to suppress the Schottky barrier and lead to a resistance drop. When the polarization is switched pointing away from NSTO, the positive screening charges will further increase the depletion and enhance the Schottky barrier. Thus, the FTJ is switched to a high-resistance state[20,21,32]. Furthermore, a shift to the negative voltage side was observed in the $R-V_p$ loops, consistent with other reported MFS-FTJs[20,21,24]. The reason is that a negative voltage

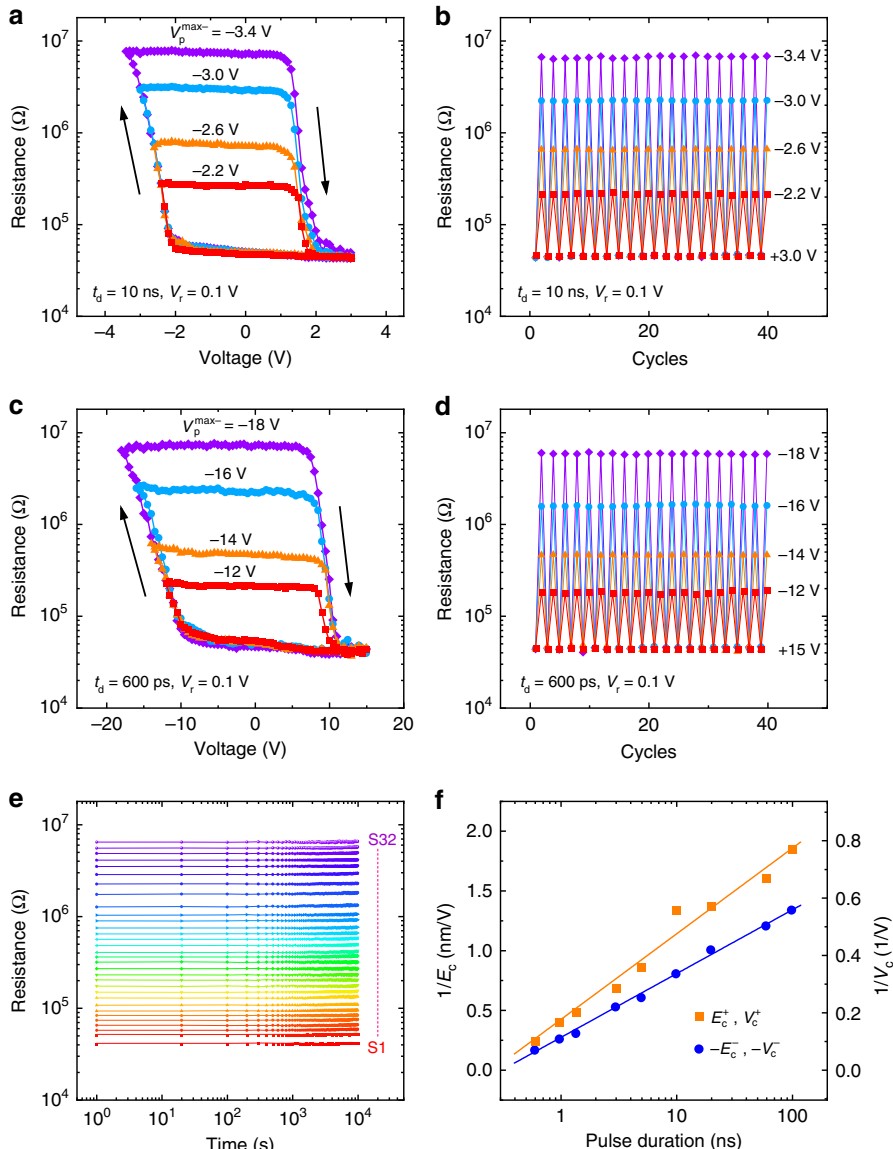

**Fig. 2 Ultrafast resistance switching at room temperature. a** Resistances measured at 0.1 V as a function of $V_p$ with $t_d = 10$ ns. **b** Typical resistance switchings between the ON state and different high-resistance states measured at 0.1 V after applying voltage pulses with $t_d = 10$ ns. **c** Resistances as a function of $V_p$ with $t_d = 600$ ps. **d** Typical resistance switchings between the ON state and different high-resistance states with $t_d = 600$ ps. The arrows in **a** and **c** represent the direction of pulse sequence. **e** Resistance retentions for 32 distinct resistive states. **f** Inverse of coercive electric field (coercive voltage) $1/E_c$ $(1/V_c)$ versus pulse duration. The solid lines are the linear fitting results.

can reversely bias the Schottky barrier, which will share more voltage than forward biased situation under a positive voltage.

It is worth noting that by varying the amplitude of $V_p^{max-}$ with $t_d = 10$ ns, the resistance can be continuously tuned from the ON state to different non-volatile intermediate resistance states, until it reaches the OFF state after a $V_p^{max-}$ of $-3.4$ V. The clear memristor behavior should be associated with the BTO ferroelectric domain partial reversal[27,28,33]. As shown in Fig. 2b, to confirm the reversible switchings among different states, cyclability measurements (Supplementary Fig. 2b) between ON state and different high-resistance states were performed by applying 10 ns pulses.

The $R$–$V_p$ loops and cyclability measurements have also been carried out at different pulse durations (600 ps $\leq t_d \leq$ 100 ns, see Fig. 2a–d and Supplementary Fig. 2c–f). The voltage required for resistance switching increases with decreasing pulse duration and reaches $+15$ V/$-18$ V for 600 ps pulses (Fig. 2c, d). The ultrafast switchings among different legible resistance states were obtained

at $t_d = 600$ ps (the current ON/OFF ratio is about $2 \times 10^2$), and the write current density is about $4 \times 10^3$ A cm$^{-2}$. To make sure that the sub-nanosecond voltage pulse is transmitted to the junction and to estimate the write current density, real-time electrical measurements were conducted during the application of the pulses (see detailed descriptions in "Methods" section and Supplementary Fig. 3). An ultralow write energy per bit $E_{write} = V_p \times I \times t_d = 500$ aJ/bit is estimated in an FTJ if the feature size is scaled to 50 nm, which is 2–3 orders of magnitude smaller than other non-volatile memories (PCRAM, MRAM, ReRAM, etc.). Furthermore, although $+15$ V/$-18$ V are required to switch the FTJ between the ON state and OFF state at 600 ps, respectively, using $+6$ V/$-8$ V we can still achieve distinguishable resistive switchings between two intermediate states with 600 ps pulses (Supplementary Fig. 4). Here, the ultrafast switching time of 600 ps in Ag/BTO/NSTO FTJs is one order of magnitude quicker than earlier reports in FTJs[14,16,25]. Compared with the SRAM,

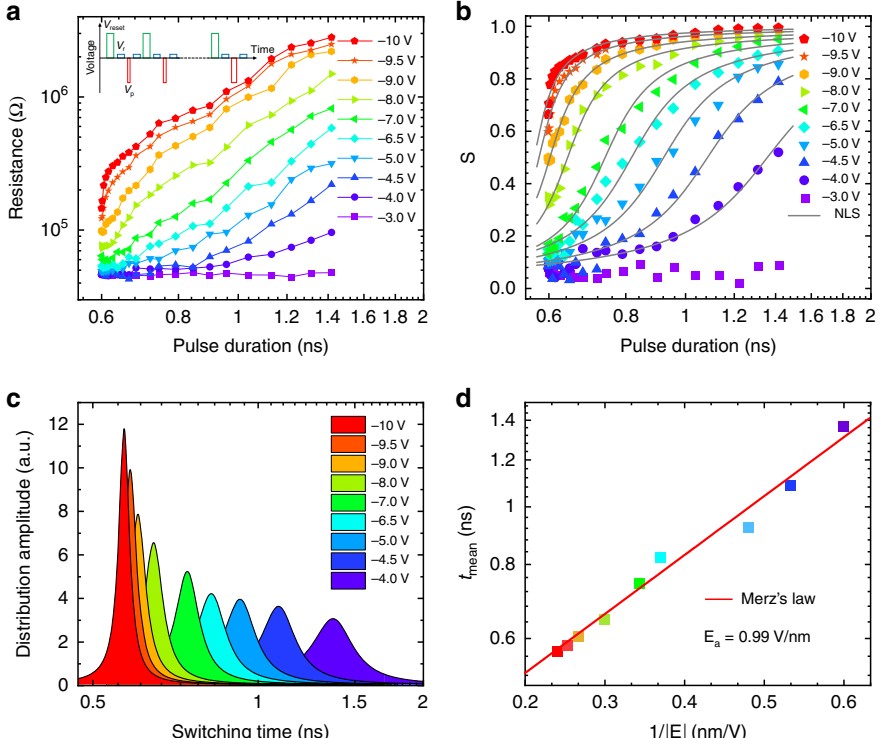

**Fig. 3 Ferroelectric domain switching dynamics. a** Resistances measured at 0.1 V and **b** relative area fraction of the ferroelectric up domain versus pulse duration. Inset of **a** illustrates the applied voltage pulse sequence. Solid curves in **b** are the fitting results by the NLS model. **c** Lorentzian distributions of switching time extracted from the fits in **b** at different pulse amplitudes. **d** Evolution of the mean switching time ($t_{mean}$) as a function of the inverse of the electric field ($1/|E|$). The solid line is the fitting result by Merz's law.

DRAM, and flash memory currently used in computers, the overall performance advantages of the FTJ memristor in ultrafast speed, ultralow energy consumption, high density, and non-volatility may be expected to overcome the storage performance gap between different levels of the memory hierarchy.

To demonstrate the capability of storing multi-memristor states, resistance retention measurements with no degradation within $10^4$ s for 32 distinct resistive states (a single cell with five bits memory capacity, obtained by varying pulse amplitudes) are displayed in Fig. 2e. The differences among any resistance states are larger than 10%, which meets the requirement in practical application[34]. This demonstrates the great potential of FTJs in high-capacity memory. In addition, the FTJs also exhibit good endurance and device uniformity, as the repeatable resistance switchings up to $10^8$–$10^9$ cycles shown in Supplementary Fig. 5a, b and the reproducible ON and OFF states in 20 different FTJs shown in Supplementary Fig. 5c.

Figure 2f shows the pulse duration-dependent coercive electric field $E_c^-$ ($E_c^+$) and coercive voltage $V_c^-$ ($V_c^+$) which are defined as the electric field and voltage, where the resistance in the $R$–$V_p$ loop reaches the middle of ON to OFF state (OFF to ON state). Accordingly, the switching time for a given electric field $E$ can be considered as the corresponding $t_d$ of the $R$–$V_p$ loop with its coercive field ($E_c$) close to $E$. The switching time is linearly proportional to $\exp(1/E)$, as the solid fitting lines in Fig. 2f. Therefore, even quicker switching time is expected with increasing the amplitude of $V_p$. It should be noted that the higher absolute value of $V_c^-$ than $V_c^+$ indicates that the switching to a high-resistance state is harder (or slower) than to a low-resistance state[24].

**Ferroelectric domain dynamics probed by junction resistance.** For further understanding the ultrafast resistive switching in the FTJ, it is necessary to study the time-dependent variations of the

FTJ resistances and the related ferroelectric domain dynamics behaviors. Thus, as illustrated in the inset of Fig. 3a, negative $V_p$ with different $t_d$ (600 ps to 1.42 ns) were used to study the switching to a high-resistance states. Here, before each setting $V_p$, a sufficiently long pulse (10 ns) $V_{reset} = +3$ V was applied to reset the FTJ to ON state. With $V_p$ changing from $-3$ to $-10$ V, the corresponding resistance evolutions are shown in Fig. 3a. It can be seen that the junction resistance switchings (i.e., the ferro-electric domain reversal) at relatively smaller $V_p$ require longer $t_d$ to achieve the same amount of resistive change.

Because of the close relationship between the ferroelectric domain configuration and the FTJ resistance $R$, the area fraction $s$ for the upward ferroelectric domains is extracted by a simple parallel circuit model as[27,33]: $1/R = (1 - s)/R_{ON} + s/R_{OFF}$. Here, the lowest ($R_{ON}$) resistance at ON state and the highest ($R_{OFF}$) resistance at OFF state were supposed to be fully downward ($s = 0$) and upward ($s = 1$) ferroelectric states. Figure 3b shows the evolution of $s$ with pulse duration at different amplitudes of $V_p$.

In FTJs, the ferroelectric switching has been successfully described by a nucleation-limited-switching (NLS) model[14,27,35] (see "Methods" section). As shown in Fig. 3b, the obtained fitting curves can reproduce the experimental data quite well. Figure 3c shows the switching time distributions, and for a larger $V_p$, a faster switching occurs in a narrower time window, consistent with previous reports[14,27]. The extracted mean switching time $t_{mean}$ also obeys Merz's law $t_{mean} \propto \exp(-E_a/E)$[14,35]. An activation electric fields $E_a$ for the ferroelectric switching of 0.99 V/nm is obtained, which is smaller than BiFeO$_3$ thin films[27], as shown in Fig. 3d. The linear relationship between switching time and $\exp(1/E)$ is consistent with the linear fitting result for $\exp(1/E_c)$ versus $t_d$ in Fig. 2f. Note that, the activation fields are intrinsic characteristics to quantitatively measure the switching of ferroelectricity, which is independent of capacitor area and measuring circuit parameters[36].

**Ultrafast ferroelectric-based synapse.** As we know, a memristor is an ideal block to emulate an artificial synapse device[28,37,38]. The ultrafast operation for artificial synapse is an effective way to enhance the computational performance of large-scale neuromorphic networks. The aforementioned memristor behavior operated at sub-nanosecond speed presents a potential application of the FTJ as an ultrafast electronic synapse device. To further demonstrate the ultrafast synapse functionality based on the ultrafast ferroelectric memristor, the STDP property, which is one of the essential learning/memory laws for emulating synaptic functions, is implemented[39,40].

Two programmable voltage spikes shown in Fig. 4a were applied to the top and bottom electrodes with a delay $\Delta t$ between the two spikes, to simulate the activities of pre-synaptic and post-synaptic neurons ($V_{pre}$ and $V_{post}$)[27,28]. The voltage amplitude of a single spike is not sufficient to induce a resistance switching. While the superimposed waveform ($V_{pre} - V_{post}$), as shown in the insets of Fig. 4b, the voltage amplitude will probably exceed the voltage threshold ($V_{th}$) to change the synaptic weight. The experimental STDP curves with the rectangular voltage pulses ($V_p$) duration of 20, 60, and 100 ns are shown in Fig. 4b, in which the device is not changed with larger $\Delta t$, whereas only closely timed spikes are able to change the synaptic weight. It is similar to the memory effect observed for two temporally close events in biological neural networks[41]. As a result, the FTJ conductance is suppressed ($\Delta G < 0$) for $\Delta t > 0$ and enhanced ($\Delta G > 0$) for $\Delta t < 0$, to emulate the synaptic depression and potentiation, respectively, indicating a typical asymmetric anti-Hebbian rule for a neuromorphic learning system[40]. The ultrafast STDP functionality of 20 ns (faster than typically reported STDP of $10^2$–$10^6$ ns[27,28]) is over three orders of magnitude faster than that of a biological synapse.

Furthermore, to emulate the synaptic weight modification with sub-nanosecond scale, the sub-nanosecond pulse (~600 ps) driven conductance change of the FTJ was measured (Supplementary Fig. 6). This implies the possible ability of the FTJs as sub-

nanosecond ultrafast synaptic devices. In the era of big data, there are more and more data need to be processed as quickly as possible, and these cannot be done by several human brains. Therefore, the neuromorphic computing needs to be much quicker than a human brain and at the same time as efficient as possible. This further highlights the advantage of the FTJs to the development of artificial intelligence.

## Discussion

Because the TER performance of the MFS-FTJ is related to the Schottky barrier at the NSTO side, it is interesting and important to investigate the role of Schottky barrier on the operation speed. Thus, Ag/BTO/NSTO FTJs with a 6 u.c. (~2.4 nm)-thick BTO barrier and various Nb concentrations from 0.05 to 0.7 wt% of NSTO were fabricated. Figure 5a, b show the $I$–$V$ curves measured at ON and OFF states, and Fig. 5c shows the FTJ currents (read at 0.1 V) and the corresponding ON/OFF ratios for different Nb concentrations. It is observed that the ON/OFF ratios change with Nb concentration, and the highest ON/OFF ratio about $3 \times 10^3$ is achieved in the FTJ with 0.1 wt% Nb, consistent with earlier report[20]. The ultrafast resistance switchings with 600 ps operation speed in the FTJs with different Nb dopants can be established at different voltage amplitudes. Figure 5d shows the pulse duration-dependent coercive voltage ($V_c^-$) which decreases with the increase of Nb concentration. This means that a faster resistive switching speed is expected with increasing the Nb-doping concentration.

To gain insight into the TER characteristics and understand the effects of Nb concentration, it is necessary to study the energy band of the MFS-FTJs. Therefore, the temperature dependent $I$–$V$ curves (from 120 to 270 K) were measured for FTJs with different Nb concentrations, and transport behaviors were analyzed by the thermally assisted tunneling mode[42–44] (see detailed descriptions in the Supplementary Figs. 7 and 8). The Schottky barrier height $\Phi_B$ and the depleted region width $W_d$ of the ON and the OFF states as a function of Nb concentration are displayed in Fig. 5e, f, respectively. Based on these parameters, the schematic energy profiles at zero bias for the ON and OFF states of the FTJs with Nb concentrations of 0.7, 0.5, 0.1, and 0.05 wt% are drawn and shown in Supplementary Fig. 9, respectively. It is obvious that $\Phi_B$ and $W_d$ in the OFF state are bigger than these in ON state, indicating the Schottky barrier are tuned by the ferroelectric polarizations. With decreasing Nb concentration, the low carrier concentration in NSTO semiconductor (Supplementary Fig. 10) leads to a more profound Schottky barrier at the BTO/NSTO interface. Therefore, the Schottky barrier with a higher barrier height and a wider depleted region width will share more voltage drop, and the partial voltage drop on BTO barrier will decrease correspondingly. As a result, a larger pulse voltage is required for a given pulse duration to flip the ferroelectric domains in BTO with a lower Nb concentration.

The Schottky barrier is not only affected by the Nb content, but also directly related to the work function of the metal electrode $\Phi_{metal}$. Therefore, the FTJs with different metal electrodes (work functions: Ag 4.26 eV, Au 5.1 eV, Pt 5.65 eV) were also fabricated and studied (Supplementary Fig. 11). The Ag/BTO/NSTO FTJ shows resistance switchings at smaller voltages, which means that the resistive switching speed will be faster at a given voltage compared with the FTJ using Pt or Au electrode. This is due to the manipulation of Schottky barrier by different metal electrodes. According to the semiconductor physics, the Schottky barrier $\Phi_B$ is proportional to the difference between $\Phi_{metal}$ and the electron affinity of the semiconductor $\chi$, namely, $\Phi_B \sim \Phi_{metal} - \chi$. A smaller $\Phi_{metal}$ will lead to a lower Schottky barrier. As a result, a smaller pulse voltage is required for a given pulse duration to

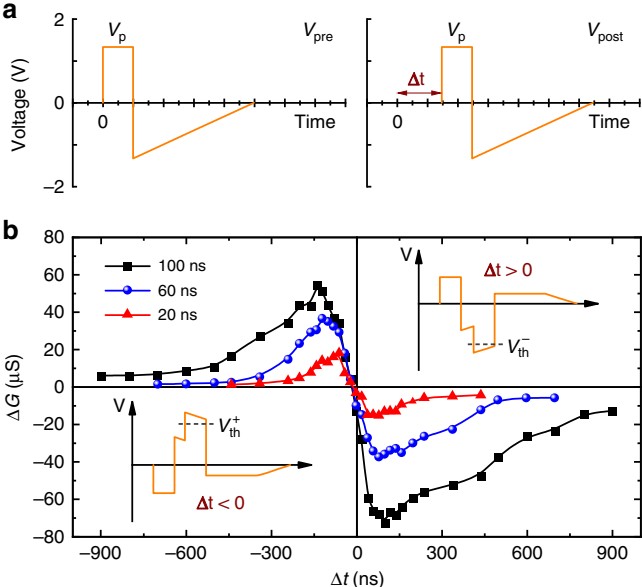

**Fig. 4 Ultrafast artificial synapses based on FTJs. a** Applied programmable voltage waveforms for pre-synaptic and post-synaptic spikes, respectively. **b** STDP measurements in the FTJ. Modulation of the FTJ conductance ($\Delta G$) as a function of the delay ($\Delta t$) between pre-synaptic and post-synaptic spikes. Insets show the waveforms produced by the superposition of the pre-synaptic and post-synaptic spikes.

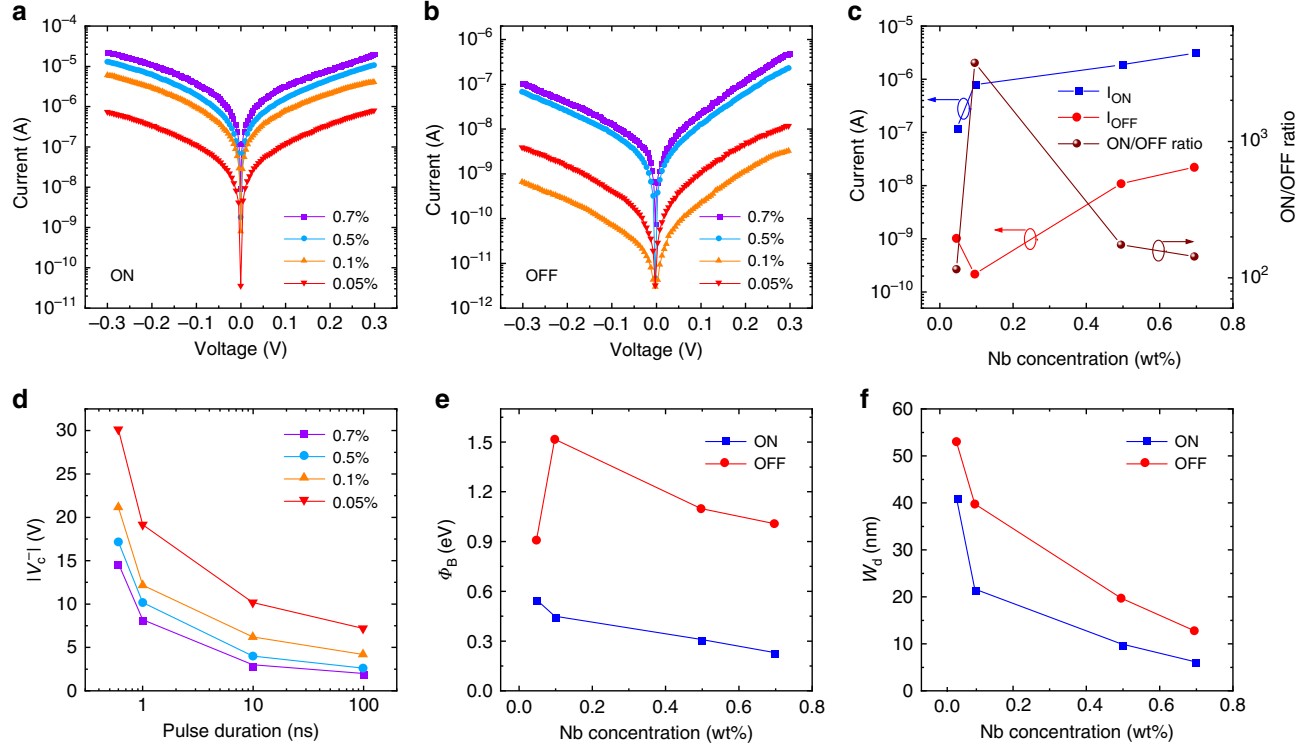

**Fig. 5 Effect of Nb-doping concentration. a, b** I–V curves of the ON and the OFF states, respectively. **c** Junction currents and ON/OFF ratios at different Nb concentrations, read at 0.1 V. **d** Absolute values of the coercive voltage $|V_c^-|$ versus the pulse duration $t_d$ with different Nb-doping concentrations. **e** Schottky barrier height ($\Phi_B$) and **f** depleted region width $W_d$ of the ON and the OFF states as a function of Nb concentration.

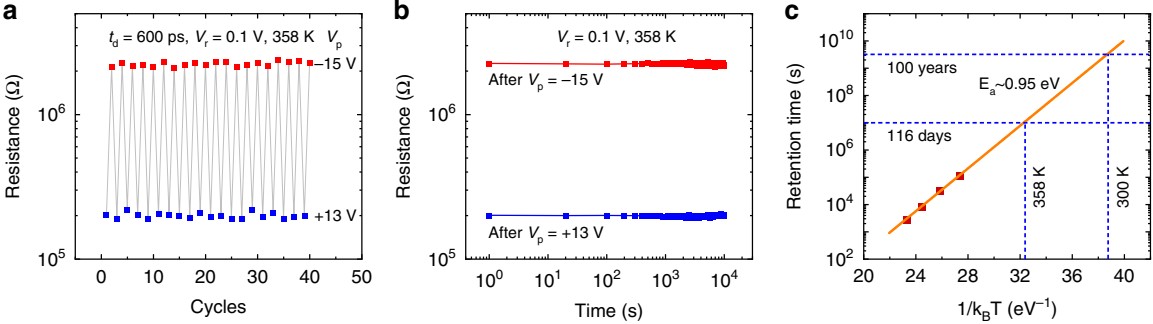

**Fig. 6 Temperature endurance of the FTJs. a** Resistance switching with $t_d = 600$ ps at 358 K. **b** Retention characteristic of the corresponding resistance states in **a**. **c** retention time versus $1/k_B T$. The points are the experimental data at 423, 448, 473 and 498 K, and the solid line is the fitting result by Arrhenius relation.

flip the ferroelectric domains in BTO with a smaller work function of metal electrode.

It is clear that, to obtain an overall excellent performance of MFS-type FTJ memories in a fast operation speed and a large ON/OFF ratio, a properly designed band structure by selecting a metal electrode with an appropriate work function and a semiconductor electrode with an appropriate carrier concentration is necessary. Based on these results, we designed and fabricated Ag/BTO/NSTO (Nb: 0.7 wt%) FTJs to achieve the sub-nanosecond resistive switching with an ON/OFF ratio of about $2 \times 10^2$, as shown in Fig. 2. It should be pointed out that although Ag ion migration can cause resistance switching, which is known as conduction bridge memristors[45], there are some experimental evidences to confirm that the resistance switchings of our Ag/BTO/NSTO FTJ are caused by ferroelectric polarization switching

rather than Ag filament conduction bridges (see detailed descriptions in the Supplementary Note 10).

To demonstrate the practical potential of the ultrafast FTJ, the ultrafast operations at temperatures higher than room temperature were investigated. The cyclability characterizations at 358 K by repeated $V_p$ with a duration of 600 ps is shown in Fig. 6a. Figure 6b shows the retention characteristic at 358 K within $10^4$ s. The retention properties at temperatures up to 498 K were also studied (Supplementary Fig. 12), and the retention time corresponding to the ON/OFF ratio of ~10 at 423, 448, 473, and 498 K is plotted in Fig. 6c as a function of $1/k_B T$, which follows an Arrhenius-type relation $t_r = t_0 \exp(E_a/k_B T)$, where $t_r$ is the retention time, $t_0$ is a constant, and $E_a$ is the activation energy. The $E_a$ extracted from the fitting is about 0.95 eV, which is comparable to that of commercial FeRAMs[46]. The room

temperature retention time, estimated by the extrapolation of retention time with temperature, can be up to 100 years. Furthermore, it is worth mentioning that the resistance switchings can still be observed well above the bulk Curie temperature (403 K)[47] of BTO. This is due to the in plane compressive strain from NSTO substrate, which stabilizes the ferroelectricity and enhances its Curie temperature, consistent with earlier reports[47,48]. Therefore, the good retention and switching endurance at an elevated temperature assert the robustness of the ultrafast operation for FTJs, meeting the requirements in integrated circuits[30].

To reveal the potential applications of the FTJs for neuromorphic computing, an ANN was simulated based on the SGD and BP algorithms[49,50], to perform an online supervised learning on the MNIST handwritten digits database (see "Methods" section and Supplementary Fig. 13). Especially, the realistic device properties including the evolution of the conductance with voltage pulses and the experimental device variations were used to build the device behavioral model for the ANN simulation (Supplementary Fig. 14). Accordingly, the effects of the cycle-to-cycle and device-to-device variations (Supplementary Fig. 15) and I–V nonlinearity (Supplementary Fig. 16) on simulated recognition accuracy were studied. Interestingly, a high recognition accuracy (>90%) can be obtained based on the above experimental performances of our FTJs, emphasizing the promising potential of FTJs as artificial synapses for constructing future neuromorphic networks.

In summary, the ultrafast resistance switchings up to 358 K of Ag/BaTiO$_3$/Nb:SrTiO$_3$ FTJs were realized using sub-nanosecond pulses. The operation speed as fast as 600 ps is achieved with the ultralow write current density about $4 \times 10^3$ A cm$^{-2}$. And the 32 distinct non-volatile resistive states in a single memory unit are also demonstrated. The overall performance advantages of FTJ memristor may be expected to solve the storage performance gap between different levels of the memory hierarchy. Furthermore, we have emulated ultrafast STDP in the FTJ memristor-based artificial synapse, and the ANN simulation using the experimental parameters suggests a high recognition accuracy of >90% on MNIST digits. It is also revealed that a higher carrier concentration of the bottom semiconductor electrode and a lower work function of the top metal electrode are beneficial for the enhancement of the operation speed for the MFS-FTJ. Our work is an experimental example demonstrating and highlighting the potential application of FTJs in ultrafast and low power consumption memristors.

## Methods

**Sample preparation**. A pulsed laser deposition technique (248 nm, 2 J cm$^{-2}$) was used to grow epitaxial BTO thin films on (001)-oriented NSTO (0.05, 0.1, 0.5, 0.7 wt% Nb) substrates at 700 °C in 10 Pa O$_2$ atmosphere. The thickness of the epitaxial BTO film is ~6 unit cells (~2.4 nm). The Ag (or Pt, or Au) top electrodes of ~70 μm in diameter and ~30 nm in thickness were sputtered on the BTO/NSTO heterostructures through a shadow mask, as schematically shown in the inset of Supplementary Fig. 1a.

**Structural and electrical characterizations**. The aberration-corrected STEM and core-level EELS (JEOL JEM-ARM200F) were used to characterize the structure of the cross-sectional Ag/BTO/NSTO. PFM measurements were carried out with conductive SCM-PIT tips (Bruker Dimension Icon). A Keithley 4200A-SCS was used to apply customized pulse waveforms for STDP measurements. The carrier concentration of NSTO and the low temperature I–V measurements were carried out in a Physical Property Measurement System (PPMS-9T, Quantum Design).

**Real-time electrical measurements**. To ensure sub-nanosecond pulses were delivered to the FTJ, we conducted a real-time electrical measurement setup similar to that in the literature[6,51–53], as shown in Supplementary Fig. 3. A pulse generator (Tektronix PSPL10300B) delivers voltage pulses with different amplitudes and durations to induce resistance switchings in the FTJs. A Keithley 2410 SourceMeter or 4200A-SCS was used to monitor the resistance change of the FTJs after applying

write voltage pulses. An oscilloscope (Tektronix DSA70804 with a bandwidth of 8 GHz) was utilized to verify the waveforms applied to the FTJs. A DC/RF switch (Radiall's RAMSES SPDT switch, 0–18 GHz) was used to separate the DC and RF circuit signals. To protect the oscilloscope against overvoltage, −10 and −6 dB attenuators are inserted before Channel 1 and Channel 2, respectively.

**Ferroelectric domain dynamics model**. The ferroelectric switching obeys the NLS model[14,27,35] with a broad Lorentzian distribution of the logarithm of switching time $t_{sw}$:

$$F(\log t_{sw}) = \frac{1}{\pi} \left( \frac{w}{(\log t_{sw} - \log t_{mean})^2 + w^2} \right) \qquad (1)$$

Here, $t_{mean}$ represents the mean switching time. $w$ and $\log t_{mean}$ are the width and center of the distribution. The normalized $s$ follows:

$$s = \int_{-\infty}^{\infty} \left\{ 1 - \exp\left[-(t_d/t_{sw})^2\right] \right\} F(\log t_{sw}) d(\log t_{sw}) \qquad (2)$$

**Neural network simulations**. A two-layer perceptron neural network with 784 input neurons, 100 hidden neurons, and 10 output neurons was simulated to implement supervised learning on the MNIST handwritten digits database, as shown in Supplementary Fig. 13. The 784 input neurons correspond to a 28 × 28 MNIST image, and the 10 output neurons correspond to 10 classes of digits (0−9). Similar to previous reports[49,50], the training algorithm based on SGD and BP algorithms, is composed of two stages: feedforward inference and feedback weight update. For each training cycle, 128 images randomly selected from 60,000 MNIST digits were set as a batch. The multilayer inference was performed layer by layer sequentially. A device behavioral model based on the realistic performances of the FTJ was created for the weight update (Supplementary Fig. 14).

## Data availability
The data supporting the findings of this study are available from the corresponding authors upon reasonable request.

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

## Acknowledgements

This work was supported by the National Natural Science Foundation of China (51790491, 51622209, 21521001, and 51972296), and National Key Research and Development Program of China (2016YFA0300103 and 2019YFA0307900), and this work was partially carried out at the USTC Center for Micro and Nanoscale Research and Fabrication.

## Author contributions

C.M. prepared the samples and performed the measurements. X.G.L. and Y.W.Y. designed and supervised the experiments and calculations. Z.L., W.C.H. and C.C.L. performed ferroelectric domain dynamics analysis and STDP measurements. Z.L., L.T.Z., X.J. carried out the ANN simulations. X.L. and Z.W.C. conducted PPMS measurements; Q.L.C. and H.Y.S. carried out the analysis of temperature-dependent transport behaviors. Y.L. carried out the HAADF-STEM and EELS measurements. X.G.L., Y.W.Y., C.M., and Z.L. wrote the manuscript. All the authors contributed to the refinement of the manuscript.

## Competing interests

The authors declare no competing interests.
