## [Peer Review File · Nature Communications]

Reviewers' comments:

Reviewer #1 (Remarks to the Author):

The manuscript by Chao Ma et al. demonstrates sub-nanosecond memristor based on the FTJ. The ultrafast switching, i.e., 540 ps pulses, as well as 32 distinct resistive states were observed in Ag/BaTiO₂/NSTO FTJ. The feasibility of even quicker switching time than 540 ps was also presented. Then, ferroelectric domain switching dynamics and STDP were demonstrated in the same structure. The TER performance related to the Schottky barrier was further examined through the modulation of the Nb doping concentration. The present work can be interesting because the authors achieved stable ultrafast switching in the FTJ.

However, since most of the experiments in the manuscript including the ferroelectric domain switching dynamics and STDP were already demonstrated by the other groups, a new finding is limited to the sub-nanosecond switching. However, I cannot find any deep discussion or any other data set for discussing about the origin of the observed ultrafast switching. Although the authors presented the effects of Nb doping concentration on the switching time, all the FTJs with different Nb doping concentrations show sub-nanosecond switching. What makes the FTJ ultrafast? Is the structure of the FTJ unique compared to the previous reports?

Although the sub-nanosecond switching was shown in the resistance switching and ferroelectric domain switching dynamics, STDP was demonstrated in tens of nanosecond scale. Did the authors try to measure STDP with sub-nanosecond scale?

Reviewer #2 (Remarks to the Author):

The authors report a memristor based on a Ag/BaTiO₃/Nb:SrTiO₃. They claim fastest operation speed (540 ps) and the highest number of states (32 states or 5 bits) per cell among all reported ferroelectric tunnel junctions (FTJs). In addition, the authors attempted to demonstrate spike-timing-dependent plasticity using their devices. Finally, they did analysis on the effect of Nb concentration of the device switching characteristics. Using FTJs to implement memristors has been reported earlier (Nat. Mater. 11, 860, 2012). Some issues need to be addressed in order for this paper to make progressive contribution to this field.

1. Ferroelectric phenomenon relies on the polarization of the material. In the past, this area has faced technical challenges such as low retention and leakage current. Part of the reason is because of the depolarization field, which is strongly related to the dielectric constant of the material. Given that complex oxides usually have high dielectric constants, the depolarization was a serious issue, as analyzed in a famous paper (IEEE EDL 23, 386, 2002). This issue was largely solved until doped HfO₂ was adopted as the ferroelectric materials, as pioneered by the Dresden group (e.g., Appl. Phys. Lett. 99, 112901, 2011). Since the current authors still used BTO as the ferroelectric layer, intrinsically the device would not have very good retention time. Could the authors comment on this?

2. The authors claimed their device a memristor, however, no IV curves with pinched hysteresis loops were provided.

3. According to the device structure, the authors used a silver top electrode and applied positive voltage on it during ON switching. As it has been widely studied that a positive voltage on silver could oxidize it into silver ions, which will then migrate to the counter electrode. This is the well-known physical picture for conduction bridge memristors. How could the authors exclude the possibility of switching from silver migration?

4. In addition to the PEM measurement, are there any TEM characterization on the devices to show the structural difference before and after the switching?

5. What is the geometry of the device including the thickness of all layers? The only info provided

is the 6 unit cell thickness of the BTO layer.

6. On the electrical measurement part, what are the retention time as a function of temperature and what is the extrapolate retention time at 85 C? Simply measuring the retention time for a short period at 85 C is not sufficient.

7. How did the authors achieve the 32 discrete resistance levels, by varying pulse number, pulse duration or pulse amplitude?

8. The sub-ns fast pulse measurement is a concern. What is the limit of the measurement system (for example, RC from the cables), and how did the authors assure the 540 ps pulses were delivered to the junction?

9. Again on the sub-ns fast pulse measurement, there are some contradictory statements regarding the switching voltages SI, "The results demonstrate that the FTJ can achieve sub-nanosecond resistive switching at less than 8 V." In main text: lines 149-150: "The voltage required for resistance switching increases with decreasing pulse duration and reaches -12 V for 540 ps pulses (Fig. 2c-d)." Lines 155-157: "Furthermore, using a voltage of about 8 V can still achieve distinguishable resistive switchings (The ON/OFF ratio about 1.3) with 540 ps pulses (Supplementary Fig. S2)."

10. Lines 24-27, the authors targeted at a 'universal memory' as one motivation of this work. However, the manuscript appears to be focused on computing, the requirements for these applications are quite different. Also, the authors definition of 'memory wall' is not right, please check original literatures and correct it.

11. Lines 67-68, the authors argued that the speed of previous synapses are not fast enough. However, being fast also means higher power consumption. In a biological system, the speed is only at ms scale so the power consumption is very low. The high computing throughput is achieved by massive parallelism in the interconnection.

12. Fig. 4b is not a representative STDP feature. The change of conductance (either in potentiation or depression) should be monotonous. However, in the results reported by the authors, the change rate of conductance drops as Δt becomes smaller. Can the authors explain this observation? Also, what were the parameters used for the experiments, such as the voltage amplitude of the pre- and post-neuron spikes? Why is the superimposition of the pulses necessary (which inevitably increases the power energy consumption)?

Reviewer #3 (Remarks to the Author):

In their manuscript entitled "Sub-nanosecond Memristor Based on Ferroelectric Tunnel Junction" Ma et al. report on the tunnel electroresistance across ultrathin films of BaTiO₃ sandwiched between Nb:SrTiO₃ and Ag electrodes. By applying voltage pulses to the ferroelectric tunnel junctions and measuring the resistance at a low dc voltage (stroboscopic measurements), they observe a hysteretic behavior of the resistance with a high resistance state for negative voltages and a low resistance state for positive voltages applied to the Ag electrode. This is consistent with previous experiments with metal/BaTiO₃/Nb:SrTiO₃ junctions in which polarization switching leads to a modulation of the Schottky barrier in Nb:SrTiO₃, a large barrier for carrier depletion (polarization away from Nb:SrTiO₃) and a low barrier for carrier accumulation (polarization toward Nb:SrTiO₃). They show that by changing the maximum write voltage they can tune the OFF-state resistance and obtain 32 clearly defined resistance states. In addition, by varying the width of the write pulses from 100 ns down to 540 ps, they observe a linear scaling of the inverse coercive field with the time in log-scale. They implement spike-timing-dependent learning rules to the ferroelectric tunnel junctions, showing the potential of these devices as artificial synapses as previously reported for BiFeO₃-based tunnel junctions. Finally, they analyze the influence of Nb-doping on the performance of the junctions and conclude that a compromise must be made between low Nb-doping (to increase the OFF/ON ratio) and fast operation (high-doping increase the operation speed). While the data are well presented in the manuscript and in the supplementary material, I suggest the authors to carefully address my questions regarding the ferroelectric characterizations of BaTiO₃, the short-voltage pulse measurements and the write

energy of the devices before recommending any publication.

1- The ferroelectric layers were deposited by pulsed laser deposition on Nb-doped SrTiO₃ substrates. Transmission electron microscopy indicates that the BaTiO₃ layer is epitaxially grown on Nb:SrTiO₃ with a *c/a* ratio of 1.05 and Ti displacements suggest a ferroelectric polarization pointing toward the substrate. Regarding the piezoresponse force microscopy (PFM) experiments, the local PFM loop in Figure 1c shows a linear amplitude with voltage which is not consistent with a ferroelectric character. Instead the amplitude should drop at the coercive voltage and saturate for both polarization directions. I suggest to remove this Figure from the manuscript.

2- In the PFM images in Figure 1d and 1e, the authors compare the signal from domains written with positive and negative voltages. They show a clear 180-degree phase contrast between domains of opposite orientation and a similar PFM amplitude for both directions. I would suggest to show the PFM signal (phase and amplitude) in the virgin state as well to corroborate the transmission electron microscopy observations, i.e. downward polarization of BaTiO₃.

3- In the geometry of the pulse-voltage experiments conducted by Ma et al., it is not clear if short voltage pulses are well transmitted across the tunnel junction or if their shape is modified when reaching the tunnel barrier. In order to make sure that the applied voltage pulse with a duration of 540ps (Figure S1a) is transmitted to the junction, the authors could conduct real-time current measurements during the application of the pulse (see Boyn et al., Appl. Phys. Lett. 113, 232902 (2018)). This would enable to measure the actual current transmitted during the voltage pulses to extract the write current densities. They should at least make a comment about it in the manuscript.

4- Page 8, line 152. How did the authors estimate their write current densities of $4 \times 10^3 \text{ Acm}^{-2}$ during the application of the 540-ps voltage pulse? See my previous point.

5- Page 9, line 196. "In FTJs, the ferroelectric switching has been successfully described by a modified Kolmogorov-Avrami-Ishibashi (KAI) model (14,33)". When mentioning "modified KAI", I believe that the authors are refereeing to the "Nucleation-limited-switching (NLS)" model. Indeed, the Lorentzian time distribution they extract in Figure 3c are the nucleation times for each electric field. This should be corrected. In addition, given the strong similarity with the paper from Boyn et al., Nature Comm. (2017) (Ref. 26), the authors should refer to this paper when presenting this model and results.

More technical details:

6- The caption of Figure 5d is wrong and must be corrected as it is not the inverse of the coercive voltage vs. pulse duration but the negative coercive voltage vs. pulse duration.

7- The details of the growth process of Ag and definition of 70-micron pads are missing.

8- The symbol for Celsius degrees does not appear in the pdf.

A list of changes

1. In **Lines 17-20 of Page 1**, the sentence “These results highlight the overall performance.....between different levels of the memory hierarchy.” was replaced by “These results may throw light on the way.....developing ultrafast neuromorphic computing systems.” in the revised manuscript.
2. In **Line 43 of Page 2 to Line 46 of Page 3**, the sentences “Based on the working principle of an FTJ¹¹.....approaching optical phonon frequencies¹⁸.” were added.
3. In **Line 68 of Page 3 to Line 74 of Page 4**, the sentences “Furthermore.....neuromorphic computational networks²⁷.” were added.
4. In **Lines 102-110 of Page 5**, the sentences “Figure 1c, d show the HAADF-STEM images.....as shown in Fig. 1e, f.” were added.
5. In **Lines 116-117 of Page 5**, the sentence “And the virgin state image of PFM for BTO shows a downward polarization, which is consistent with the STEM results in Fig. 1a.” was added.
6. In **Lines 131-133 of Page 6**, the sentence “The representative I - V curves.....with a memristive characteristic (Supplementary Fig. S1).” was added.
7. In **Lines 173-176 of Page 9**, the sentence “To make sure that the sub-nanosecond voltage pulse..... (see detailed descriptions in Methods and Supplementary Fig. S3).” was added.
8. In **Lines 179-182 of Page 9**, the sentence “Furthermore, although +15 V/-18 V are required.....between two intermediate states with 600 ps pulses (Supplementary Fig. S4).” was added.
9. In **Lines 187-188 of Page 9**, the phrase “the memory wall issue” was replaced by “the storage performance gap between different levels of the memory hierarchy”
10. In **Line 191 of Page 9**, the phrase “obtained by varying pulse amplitudes” was added in the revised manuscript.
11. In **Lines 220, 231 of Page 11** and **Line 404 of Page 19**, the “modified Kolmogorov-Avrami-Ishibashi (KAI) model” was replaced by “nucleation-limited-switching (NLS) model”.
12. In **Lines 273-280 of Page 14**, the sentences “Furthermore.....the development of artificial intelligence.” were added.
13. In **Line 319 of Page 16 to Line 342 of Page 17**, the sentences “In addition, the

FTJs with different metal electrodes..... rather than Ag filament conduction bridges (see detailed descriptions in the Supplementary S10).” were added.

14. In **Lines 347-358 of Page 17**, the sentences “The retention properties at temperatures up to 498 Kconsistent with earlier reports^{47,48}.” were added.
15. In **Line 381 of Page 18 to Line 383 of Page 19**, the sentences “The thickness of the epitaxial BTO film..... through a shadow mask” were added.
16. In **Lines 392-402 of Page 19**, the sentences “**Real-time electrical measurements**..... Channel 1 and Channel 2, respectively.” were added.
17. In the revised manuscript, **Figs. 1, 2, 3 and 6** were replotted.
18. In the Supplementary information, **S1, S3, S6, S9, S10, and S11** were added.
19. In **S4** of Supplementary information, the descriptions were rewritten.

There are also some minor revisions about the sentences to improve English language as well as several reference updates, which are not listed here.

Responses to Reviewer #1

Thank you very much for your valuable comments on our manuscript. We have made careful revisions according to your comments. We wish that the revised version and the responses would be satisfactory to you.

Comment 1. *However, since most of the experiments in the manuscript including the ferroelectric domain switching dynamics and STDP were already demonstrated by the other groups, a new finding is limited to the sub-nanosecond switching. However, I cannot find any deep discussion or any other data set for discussing about the origin of the observed ultrafast switching. Although the authors presented the effects of Nb doping concentration on the switching time, all the FTJs with different Nb doping concentrations show sub-nanosecond switching. What makes the FTJ ultrafast? Is the structure of the FTJ unique compared to the previous reports?*

Answer 1: We thank the reviewer for the question and great suggestions. As you pointed out, the sub-nanosecond switching in FTJs is one of the major breakthroughs achieved in our manuscript. The operation speed at sub-nanosecond, capable of following up with a commercial CPU, is a performance milestone for non-volatile memories. In our manuscript, the demonstrations of sub-nanosecond ultrafast-speed, high density, and ultralow energy consumption make FTJ devices to be one of the most promising next-generation memories to meet the ever high requirements for fast and efficient big data processing.

Based on the working principle of an FTJ¹, its switching time is mainly determined by the ferroelectric domain switching dynamics. In principle, the ferroelectric polarization reversal may theoretically be as fast as around 10^{-13} s, approaching optical phonon frequencies². However, for an FTJ with an ultrathin ferroelectric film, many factors become significantly important in affecting ferroelectric switching and thus influence the resistive switching speed of the FTJ, such as electrode effect, interface barriers, depolarization field, domain patterns, size and strain effects, and so on³. There is still no report showing whether FTJs could be operated at sub-nanosecond.

Previous reports have demonstrated that, owing to the ferroelectricity tuned interfacial Schottky barrier, Pt/BTO/NSTO metal/ferroelectric/semiconductor (MFS)-type FTJs show high ON/OFF ratios^{4,5}. However, it is still not clear how the magnitude of the Schottky barrier affects the switching speed, the operation voltage, *etc.*

To realize an FTJ with ultrafast speed at sub-nanosecond scale, high ON/OFF ratio ($>10^2$), multi-level storage capability, and applicable operation voltage (for comparison, voltages of 12-20 V are used to write and erase NAND flash memory cells⁶) simultaneously, proper material and structure designs have been made as follows.

Fig. RI-1 | FTJs with different metal electrodes. **a** Energy profiles of the separated metal, BTO, and NSTO, where Φ_{metal} is the work function of metal (Ag 4.26 eV, Au 5.1 eV, Pt 5.65 eV), $\chi_{BTO} = 3.9$ eV is the electron affinity of BTO, $\chi_{NSTO} = 4.0$ eV is the electron affinity of NSTO, E_{vac} is the vacuum level, and E_C , E_V , E_F are the conduction band minimum, the valence band maximum, and the Fermi level of NSTO, respectively. **b** Resistances measured at 0.1 V *versus* pulse amplitude V_p with $t_d = 100$ ns for Ag/BTO/NSTO, Au/BTO/NSTO, and Pt/BTO/NSTO FTJs with a 6 u.c.-thick BTO barrier and 0.7wt% Nb concentration. The arrows indicate the direction of pulse sequence. Energy profiles of **c** Ag/BTO/NSTO, **d** Au/BTO/NSTO, and **e** Pt/BTO/NSTO FTJs.

1) Several metals with different work functions (Ag 4.26 eV, Au 5.1 eV, Pt 5.65 eV) were used as top electrodes to tune the Schottky barrier, which has not been

investigated in MFS-type FTJs previously. According to the semiconductor physics, the Schottky barrier V_{bi} is proportional to the difference between the metal work function Φ_{metal} (the energy difference between the metal Fermi level and the vacuum level) and the electron affinity of the semiconductor χ (the difference between the semiconductor conduction band edge and the vacuum level), namely, $V_{bi} \sim \Phi_{\text{metal}} - \chi$, as shown in Fig. RI-1a. A smaller work function of the metal electrode will lead to a lower Schottky barrier. Because this Schottky barrier will share a considerable voltage drop from the total applied voltage, lowering the Schottky barrier will increase the partial voltage drop across the BTO barrier. It means that a smaller pulse voltage could flip the ferroelectric domains of BTO in FTJs using a metal electrode with a smaller work function.

Figure RI-1b shows R - V_p loops ($t_d = 100$ ns) of the MFS-type FTJs with Ag, Au, and Pt metal electrodes. The energy profiles of these FTJs are schematically shown in Fig. RI-1c-e. It can be seen that the Pt/BTO/NSTO FTJ shows the biggest ON/OFF ratio. This may be one of the reasons why previous researchers mostly used Pt with a high work function as an electrode for their FTJs^{4,5}. However, the large Schottky barrier in the Pt/BTO/NSTO FTJ will result in a high operation voltage, which may not be beneficial to practical applications in high operating speed. The utilization of the Ag electrode, by contrast, can greatly reduce the operation voltage, which means that the resistive switching speed will be much faster at a given voltage. Furthermore, it is worth mentioning that the Ag/BTO/NSTO FTJ still presents an ON/OFF ratio as high as about 2×10^2 , which is enough for non-volatile memories with 32 states (Fig. 2).

2) Semiconducting NSTO electrodes with various carrier concentrations were used to further optimize the band structure of the Ag/BTO/NSTO FTJs for sub-nanosecond switchings. It is clear that the higher Nb concentration of the NSTO leads to smaller operation voltages and faster operation speeds, as shown in Fig. 5 (see the details in the **Discussion** section of the revised manuscript).

In summary, to obtain an overall excellent performance MFS-type FTJ memories in the fast operation speed and the large ON/OFF ratio, a properly designed band structure by selecting a top metal electrode with an appropriate work function and a bottom semiconductor electrode with an appropriate carrier concentration is necessary. Based on these results, we designed and fabricated Ag/BTO/NSTO (Nb: 0.7wt%) FTJs

to achieve the sub-nanosecond resistive switching with an ON/OFF ratio of about 2×10^2 .

Figure RI-1 is added as **Supplementary Fig. S11**. The corresponding discussions were added in **Line 319 of Page 16 to Line 338 of Page 17** of the revised manuscript and **S9** of the revised Supplementary Information.

Comment 2. *Although the sub-nanosecond switching was shown in the resistance switching and ferroelectric domain switching dynamics, STDP was demonstrated in tens of nanosecond scale. Did the authors try to measure STDP with sub-nanosecond scale?*

Answer 2: This is a very good suggestion. In fact, the memristor behavior operated at sub-nanosecond speed shown in Fig. 2 indicates the capability of the FTJ to implement the sub-nanosecond STDP synaptic simulation. However, because of the limitation of instruments, we are unable to measure the STDP with sub-nanosecond scale at present. In the STDP measurements, two programmable voltage spikes shown in Fig. 4a were applied to the top and bottom electrodes with a delay time Δt between the two spikes, to simulate the activities of biological pre- and post-synaptic neurons (V_{pre} and V_{post})^{7,8}. Therefore, for realizing STDP measurements with sub-nanosecond scale, the instruments must be able to generate sub-nanosecond *programmable* waveforms and control the delay time Δt to the sub-nanosecond scale. Unfortunately, the Keithley 4200A-SCS we used for STDP measurements can only reach a minimum time scale of 20 ns. In addition, although our Tektronix PSPL10300B can generate a sub-nanosecond voltage pulse, the waveform of this pulse is not programmable. It would be very interesting to perform the STDP measurements with sub-nanosecond scale, and related researches will be conducted in future.

In spite of that mentioned above, to emulate the synaptic weight modification with sub-nanosecond scale, the sub-nanosecond pulse (~600 ps) driven conductance change of the FTJ was measured, as shown in Fig. RI-2. The conductance of the memristor can be manipulated gradually by increasing the amplitude (in a step of 0.5 V) of negative or positive voltage pulses, representing the depression or potentiation of the synaptic weight. In other words, it demonstrates the possible ability of the FTJs as sub-nanosecond ultrafast synaptic devices.

Figure RI-2 is added as **Supplementary Fig. S6**. The corresponding discussions were added in **Lines 273-276 of Page 14** of the revised manuscript and **S6** of the revised Supplementary Information.

Fig. RI-2 | Sub-nanosecond pulse driven synaptic weight modulation. The negative (0 to -18 V) and positive pulses (0 to +15 V) with a step of 0.5 V were applied to the FTJ. The pulse duration is ~600 ps, and the device resistance was read at a bias of 0.1V.

References

1. Garcia, V. & Bibes, M. Ferroelectric tunnel junctions for information storage and processing. *Nat. Commun.* **5**, 4289 (2014).
2. Jiang, A. Q., Lee, H. J., Hwang, C. S. & Scott, J. F. Sub-Picosecond Processes of Ferroelectric Domain Switching from Field and Temperature Experiments. *Adv. Funct. Mater.* **22**, 192-199 (2012).
3. Fridkin, V. M. & Ducharme, S. *Ferroelectricity at the Nanoscale: Basics and Applications*. (Springer, 2014).
4. Wen, Z., Li, C., Wu, D., Li, A. D. & Ming, N. B. Ferroelectric-field-effect-enhanced electroresistance in metal/ferroelectric/semiconductor tunnel junctions. *Nat. Mater.* **12**, 617 (2013).
5. Xi, Z. N., Ruan, J. J., Li, C., Zheng, C. Y., Wen, Z., Dai, J. Y., Li, A. D. & Wu, D. Giant tunnelling electroresistance in metal/ferroelectric/semiconductor tunnel junctions by engineering the Schottky barrier. *Nat. Commun.* **8**, 15217 (2017).
6. Chanthbouala, A., Crassous, A., Garcia, V., Bouzehouane, K., Fusil, S., Moya, X., Allibe, J., Dlubak, B., Grollier, J., Xavier, S., Deranlot, C., Moshar, A., Proksch, R., Mathur, N. D., Manuel, B. & Barthélemy, A. Solid-state memories based on

- ferroelectric tunnel junctions. *Nat. Nanotechnol.* **7**, 101 (2012).
7. Boyn, S., Grollier, J., Lecerf, G., Xu, B., Locatelli, N., Fusil, S., Girod, S., Carretero, C., Garcia, K., Xavier, S., Tomas, J., Bellaiche, L., Bibes, M., Barthelemy, A., Saighi, S. & Garcia, V. Learning through ferroelectric domain dynamics in solid-state synapses. *Nat. Commun.* **8**, 14736 (2017).
 8. Kuzum, D., Yu, S. M. & Wong, H.-S. P. Synaptic electronics: materials, devices and applications. *Nanotechnology* **24**, 382001 (2013).

Responses to Reviewer #2

Thank you very much for your valuable comments and suggestions on our manuscript. We have carefully revised the manuscript following your suggestions point by point as follows:

Comment 1. *Ferroelectric phenomenon relies on the polarization of the material. In the past, this area has faced technical challenges such as low retention and leakage current. Part of the reason is because of the depolarization field, which is strongly related to the dielectric constant of the material. Given that complex oxides usually have high dielectric constants, the depolarization was a serious issue, as analyzed in a famous paper (IEEE EDL 23, 386, 2002). This issue was largely solved until doped HfO₂ was adopted as the ferroelectric materials, as pioneered by the Dresden group (e.g., Appl. Phys. Lett. 99, 112901, 2011). Since the current authors still used BTO as the ferroelectric layer, intrinsically the device would not have very good retention time. Could the authors comment on this?*

Answer 1: We thank the reviewer for raising this important topic. HfO₂-based ferroelectrics offer many advantages including low process temperature (~450 °C), good CMOS-compatible process, *etc.* So it is one of the promising ferroelectric materials for next-generation non-volatile memories, and we will investigate the ultrafast resistance switching of HfO₂-based FTJs in the future. Furthermore, as demonstrated in the paper you mentioned (Müller, J., *et al.*, Appl. Phys. Lett. 99, 112901, 2011)¹, the authors obtained a better read out of the polarization state in Hf_{0.5}Zr_{0.5}O₂ based charge-type FeRAM due to the low dielectric constant of Hf_{0.5}Zr_{0.5}O₂. Different from an FeRAM in which the ferroelectric state is read out destructively by the ferroelectric switching current, the ferroelectric state in FTJs is read out non-destructively by the ferroelectric polarization orientation-dependent resistance.

As you pointed out, the research field of ferroelectric memory (more specifically, ferroelectric field transistor FET based memories) has faced technical challenges such as low retention and leakage current. The depolarization field is an important factor for decreasing the stability of ferroelectric polarization, and it will be smaller for a ferroelectric material with a higher dielectric constant. For example, in the first paper you mentioned (Ma, T. P. and Jin-Ping Han, IEEE EDL 23, 386, 2002)², the depolarization field (E_{dp}) is described by

$$E_{dp} = PC_F [\varepsilon(C_{IS} + C_F)]^{-1} = P \left[\varepsilon \left(\frac{C_{IS}}{C_F} + 1 \right) \right]^{-1} = P \left(\frac{C_{IS}}{S/d} + \varepsilon \right)^{-1} \quad (\text{R1})$$

where P , C_F , ε , S , and d are the polarization, capacitance, dielectric constant, area, and thickness of the ferroelectric film, respectively, and C_{IS} is the semiconductor capacitance which may be generalized to represent the series combination of an insulating buffer on top of the semiconductor. It can be seen that a higher dielectric constant ε will lead to a smaller depolarization field E_{dp} . The same conclusion is also demonstrated by Kim, D. J., *et al.* (*Phys. Rev. Lett.* 95, 237602, 2005)³, in which the depolarization field is given by

$$E_{dp} = -\frac{P}{\varepsilon_0 \varepsilon_F} \left(\frac{2\varepsilon_F/d}{2\varepsilon_F/d + \varepsilon_e/\lambda} \right) = -\frac{2P}{\varepsilon_0 (2\varepsilon_F + \varepsilon_e d/\lambda)}. \quad (\text{R2})$$

ε_F and ε_e are the relative dielectric constants of the ferroelectric layer and the electrode, respectively. λ is the screening length in electrodes. Eq. R2 also suggests that a higher dielectric constant of the ferroelectric layer may be helpful to reduce the depolarization field.

To reveal the retention time of our BTO based FTJs, we investigated the temperature dependences of retention properties of the Ag/BTO/NSTO FTJ, as shown in Fig. RII-1a. The retention time corresponding to the ON/OFF ratio of ~ 10 at 423, 448, 473 and 498 K is plotted in Fig. RII-1b as a function of $1/k_B T$, which follows an Arrhenius-type relation $\tau = \tau_0 \exp(E_a/k_B T)$, where τ is the retention time, τ_0 is a constant, and E_a is the activation energy. The E_a extracted from the fitting is about 0.95 eV, which is comparable to that of commercial FeRAMs⁴. The room temperature retention time of the Ag/BTO/NSTO is estimated by the extrapolation of retention time with temperature, which can be up to 100 years, similar to that reported by Xi *et al.*⁵. Thus, it meets the requirement of practical applications for the non-volatile memory devices. Furthermore, it is worth mentioning that the resistance switchings can still be observed well above the bulk Curie temperature (403 K)⁶ of BTO. This is due to the in plane compressive strain from NSTO substrate which stabilizes the out-of-plane ferroelectricity of BTO and enhances its Curie temperature, consistent with earlier reports^{5,6}.

Figure RII-1a is added as **Supplementary Fig. S12**, and Fig. RII-1b is added as **Fig. 6c**. The corresponding discussions were added in **Lines 347-358 of Page 17** of the revised manuscript and **S11** of the revised Supplementary Information.

Fig. RII-1 | High-temperature retention properties. **a** Retention properties of the Ag/BTO/NSTO FTJ at 423, 448, 473 and 498 K. **b** retention time *versus* $1/k_B T$. The solid line is the fitting result by using the Arrhenius relation $\tau = \tau_0 \exp(E_a/k_B T)$.

Comment 2. *The authors claimed their device a memristor, however, no IV curves with pinched hysteresis loops were provided.*

Fig. RII-2 | I-V loops of the Ag/BTO/NSTO FTJ. **a, b** I-V curves of a Ag/BTO/NSTO FTJ measured by sweeping the voltage from 0 to 2 V, then 2 V to -2 V, and finally back to 0 V. The arrows indicate the voltage sweeping direction.

Answer 2: We are very grateful for the reviewer's great suggestion. Following your suggestion, the representative I-V curves of the FTJs are shown in Fig. RII-2. It can be seen that typical pinched hysteresis loops observed show the memristive characteristic, similar to the previous report⁷. In addition, the I-V curves reveal a rectifying transport character, indicating the existence of the Schottky barrier for MFS-type FTJs.

Figure RII-2 is added as **Supplementary Fig. S1**. The experimental results and relevant description have been added in **Lines 131-133 of Page 6** of the revised manuscript and the **S1** of the revised Supplementary information.

Comment 3. *According to the device structure, the authors used a silver top electrode and applied positive voltage on it during ON switching. As it has been widely studied that a positive voltage on silver could oxidize it into silver ions, which will then migrate to the counter electrode. This is the well-known physical picture for conduction bridge memristors. How could the authors exclude the possibility of switching from silver migration?*

Answer 3: Thank you for pointing out this important issue. In our case, there are some experimental evidences to exclude the occurrence of Ag migration.

1) Figure RII-3a, b show the HAADF-STEM images from the Ag/BTO/NSTO FTJs at OFF and ON states by applying -3 V and +3 V voltage pulses ($t_d = 100$ ns), respectively. The upward and downward displacements of Ti ions are observed at OFF and ON states, consistent with the ferroelectric resistive switching mechanism. It should be noted that there is no Ag migration or Ag filament in BTO, as shown in Fig. RII-3c, d. This is a direct evidence to exclude the occurrence of Ag migration.

Fig. RII-3 | Structural characterizations of the Ag/BTO/NSTO FTJs. **a, b** HAADF-STEM images of the Ag/BTO/NSTO FTJs at OFF state and ON state with the insets showing upward and downward displacements of Ti ions, respectively. The orange and green spheres denote Ba and Ti ions, respectively. **c, d** Ag element distributions at OFF and ON states measured by the EDS mapping.

2) The resistance switching characteristics of the FTJs are different from the Ag migration based resistance switchings. The I - V characteristics at ON states for the FTJs are non-linear, following the thermally-assisted tunneling model (Supplementary Fig. S7). These are different from the typical linear I - V curves for conduction bridge memories based on Ag filaments at ON states⁸.

3) For understanding the ultrafast resistive switching in the FTJ, we have studied the time-dependent variation of the FTJ resistance and the related ferroelectric domain dynamics behaviors. As shown in Fig. 3, it can be seen that the resistive switching of the Ag/BTO/NSTO FTJ is closely correlated with a nucleation-limited-switching (NLS) model of the ferroelectric domain dynamics^{9,10}.

4) As shown in Fig. RII-4, not only the FTJ with Ag electrode, but also the FTJs with Au and Pt electrodes show the resistance switching effects. This is also one of the evidences to exclude the occurrence of Ag migration.

Fig. RII-4 | FTJs with different metal electrodes. Resistances measured at 0.1 V *versus* pulse amplitude V_p with $t_d = 100$ ns for Ag/BTO/NSTO, Au/BTO/NSTO, and Pt/BTO/NSTO FTJs with a 6 u.c.-thick BTO barrier and 0.7wt% Nb concentration. The arrows indicate the direction of pulse sequence.

All the above experimental results confirm that the resistance switching of the Ag/BTO/NSTO FTJ is caused by the ferroelectric polarization switching rather than the conduction bridge based on Ag filaments.

Figure RII-3 is added as **Fig. 1c-f**. Figure RII-4 is added as **Supplementary Fig. S11b**. The experimental results and relevant descriptions have been added in **Lines 338-342 of Page 17** of the revised manuscript and **S9** and **S10** of the revised Supplementary information.

Comment 4. *In addition to the PFM measurement, are there any TEM characterization on the devices to show the structural difference before and after the switching?*

Answer 4: Following your suggestion, TEM measurements for FTJs before and after the switchings were carried out. As shown in Fig. 1a, the virgin state of the FTJ before the switching shows downward displacements of Ti ions, indicating a downward polarization in BTO. Figure RII-3a, b show the HAADF-STEM images from the Ag/BTO/NSTO FTJs after poling the ferroelectricity upward and downward by -3 V and +3 V ($t_d = 100$ ns), respectively. As shown in the inset of Fig. RII-3a, b, the Ti ion displacements of about 18 pm and -17 pm along the [001] direction suggest an upward and downward polarized BTO, respectively, which is consistent with the ferroelectric polarizations.

Figure RII-3a, b is added as **Fig. 1c, d**. The experimental results and relevant description have been added in **Lines 102-110 of Page 5** of the revised manuscript.

Comment 5. *What is the geometry of the device including the thickness of all layers? The only info provided is the 6 unit cell thickness of the BTO layer.*

Answer 5: The MFS-type FTJ memristors is constructed based on Ag/BTO/NSTO heterostructures. The (001) oriented single-crystalline NSTO ($5 \times 5 \times 0.5$ mm) is chosen as the semiconductor substrate. The thickness of the epitaxial BTO film is ~6 unit cells. Ag top electrodes of 70 μ m in diameter and 30 nm in thickness were sputtered on the BTO/NSTO heterostructures through a shadow mask.

The relevant descriptions have been added in **Line 381 of Page 18 to Line 383 of Page 19** of the **Methods** section of the revised manuscript.

Comment 6. *On the electrical measurement part, what are the retention time as a function of temperature and what is the extrapolate retention time at 85 °C? Simply measuring the retention time for a short period at 85 °C is not sufficient.*

Answer 6: Following your advice, we investigated the high-temperature retention properties of the Ag/BTO/NSTO FTJ, as shown in Fig. RII-1, which is discussed in detail in the response to your Comment 1. The 358 K (85°C) retention time of the Ag/BTO/NSTO is estimated by the extrapolation of retention time with temperature, which can be up to 116 days. And room temperature retention time is expected to be

about 100 years, similar to that reported by Xi *et al*⁵. So it meets the requirement of practical applications for the non-volatile memory devices².

Figure RII-1a is added as **Supplementary Fig. S12**, and Fig. RII-1b is added as **Fig. 6c**. The corresponding discussions were added in **Lines 347-358 of Page 17** of the revised manuscript and **S11** of the revised Supplementary Information.

Comment 7. *How did the authors achieve the 32 discrete resistance levels, by varying pulse number, pulse duration or pulse amplitude?*

Answer 7: The 32 distinct resistive states of the FTJ were obtained by varying pulse amplitudes. The relevant descriptions were added in **Line 191 of Page 9** of the revised manuscript.

Comment 8. *The sub-ns fast pulse measurement is a concern. What is the limit of the measurement system (for example, RC from the cables), and how did the authors assure the 540 ps pulses were delivered to the junction?*

Answer 8: Many thanks to the reviewer for the good question. To ensure that sub-nanosecond pulses were delivered to the FTJ, we conducted a real-time electrical measurement setup similar to that in the literature¹¹⁻¹³, as shown in Fig. RII-5a. A pulse generator (Tektronix PSPL10300B with the shortest pulse duration about 540 ps) delivers voltage pulses with different amplitudes and durations to induce resistance switchings in the FTJs. A Keithley 2410 SourceMeter was used to monitor the resistance change of the FTJs after applying write voltage pulses. An oscilloscope (Tektronix DSA70804 with a bandwidth of 8 GHz) was utilized to verify the waveforms applied to the FTJs. A DC/RF switch (Radiall's RAMSES SPDT switch, 0-18 GHz) was used to separate the DC and RF circuit signals. To protect the oscilloscope against overvoltage, -10 dB and -6 dB attenuators are connected before Channel 1 and Channel 2, respectively.

In this way, we can record the voltage pulse applied to the FTJ top electrode using Channel 1 of the oscilloscope. The signal transmitted through the FTJ is also recorded by the oscilloscope Channel 2. For example, Fig. RII-5b shows that a voltage pulse of 540 ps in duration and 15 V in amplitude was successfully applied to the FTJ. While the signal transmitted through the FTJ shows a pulse duration of about 600 ps, as shown in Fig. RII-5c. In other words, the RC delay τ_{RC} extends the 540 ps pulse to 600 ps, and τ_{RC} is estimated to be about $(600-540)/2 = 30$ ps. Compared with the sub-nanosecond pulse signal, such a small RC delay would not affect the conclusions obviously.

According to the results mentioned above, we changed the sub-nanosecond resistive switching time from 540 ps to 600 ps in the revised manuscript.

Figure RII-5 is added as **Supplementary Fig. S3**. The experimental results and relevant descriptions have been added in **Lines 392-402 of Page 19** of the **Methods** section of the revised manuscript and the **S3** of the revised Supplementary information.

Fig. RII-5 | Real-time electrical measurements. **a** Schematic description of the real-time electrical measurement setup. **b** Voltage pulse of 540 ps in duration and 15 V in amplitude applied to the FTJ top electrode. **c** Signal transmitted through the FTJ with a duration of 600 ps.

Comment 9. *Again on the sub-ns fast pulse measurement, there are some contradictory statements regarding the switching voltages SI, “The results demonstrate that the FTJ can achieve sub-nanosecond resistive switching at less than 8 V.” In main text: lines 149-150: “The voltage required for resistance switching increases with decreasing pulse 150 duration and reaches -12 V for 540 ps pulses (Fig. 2c-d).” Lines 155-157: “Furthermore, using a voltage of about 8 V can still achieve distinguishable resistive switchings (The ON/OFF ratio about 1.3) with 540 ps pulses (Supplementary Fig. S2).”*

Answer 9: We apologize for our misleading statements. As demonstrated by the $R-V_p$ loops in Fig. 2c, the resistance of the FTJ changes smoothly between the ON state ($\sim 4 \times 10^4 \Omega$) and the OFF state ($\sim 8 \times 10^6 \Omega$) with the applications of 600 ps voltage pulses, and there are many intermediate resistive states between ON and OFF states. Although +15 V/-18 V at 600 ps are needed to switch the FTJ to the ON state and the OFF state (Fig. 2d), respectively, resistive switchings between intermediate states require smaller voltages, as shown in Supplementary Fig. S4. Using +6 V/-8 V (Supplementary Fig. S4d) we can still achieve distinguishable resistive switchings between two intermediate states ($\sim 2.2 \times 10^5 \Omega$ and $\sim 3 \times 10^5 \Omega$).

The relevant descriptions have been added in **Lines 179-182 of Page 9** of the revised manuscript and the **S4** of the revised Supplementary information.

Comment 10. *Lines 24-27, the authors targeted at a ‘universal memory’ as one motivation of this work. However, the manuscript appears to be focused on computing, the requirements for these applications are quite different. Also, the authors definition of ‘memory wall’ is not right, please check original literatures and correct it.*

Answer 10: Thank you for raising this important topic. Actually, with many excellent and balanced performances, our FTJ based memristor may act as not only a memory device targeting at “universal memory” but also an artificial synapse device for computing. On the one hand, the FTJ as a non-volatile memory is expected to become a “universal memory” which includes the best attributes of all different commercial memories (static RAM, dynamic RAM, and flash) — high speed, low energy consumption, high endurance, high density, non-volatility¹⁴. On the other hand, the FTJ possesses continuous tunable resistances in one unit cell, and thus can serve as an artificial synapse with ultrafast synaptic emulation, such as STDP. This demonstrates its ability in constructing neuromorphic computing networks. Therefore, we target at the “universal memory” and neuromorphic computing as both motivations in our work. We apologize that we didn’t state this clearly in our previous manuscript.

Thanks for pointing out the confusing definition of the “memory wall”. In the von Neumann architecture, there exists wide performance gaps between the central processing unit (where the data is processed) and the computer memory (where data is stored)^{15,16} as well as between the fast-accessing memory and long-term storage¹⁷. The former gap is named “memory wall”^{15,16}. To clarify these clearer, we used the phrase

“storage performance gap between different levels of the memory hierarchy” to replace the “memory wall”.

The relevant description has been added in the **Introduction** section of the revised manuscript.

Comment 11. *Lines 67-68, the authors argued that the speed of previous synapses are not fast enough. However, being fast also means higher power consumption. In a biological system, the speed is only at ms scale so the power consumption is very low. The high computing throughput is achieved by massive parallelism in the interconnection.*

Answer 11: Thank you for raising this important topic. As you pointed out, the human brain can easily process complex tasks such as a visual processing by using millisecond scale spikes and an average frequency of 10 Hz¹⁸. The operating speed of the brain is suitable to maintain low power consumption. This is the reason why we would like to setup the neuromorphic computing. However, this doesn't mean the neuromorphic computing has to work at a low frequency. In the era of big data, there are more and more data need to be processed as quickly as possible. For example, there are tons of pictures and videos need to be analyzed daily which cannot be done by several human brains. Therefore, the neuromorphic computing needs to work much quicker than human brain.

As for the power consumption, a faster speed will lead to a higher power consumption, and a tradeoff between the operation speed and the power consumption should always be carefully considered¹⁸. However, considering the same amount of computing tasks, a higher power consumption does not mean a higher energy consumption. Actually, a faster operation speed is helpful to reduce the synaptic energy consumption per programming E , which can be estimated by $E = VIt$, where V is the pulse amplitude, I is the current flowing across the device, and t is the programming pulse duration. In order to minimize synaptic energy consumption all three components need to be minimized. Therefore, low programming energy may be obtained by minimizing programming time.

The relevant descriptions have been added in **Lines 276-280 of Page 14** of the revised manuscript.

Comment 12. *Fig. 4b is not a representative STDP feature. The change of conductance (either in potentiation or depression) should be monotonous. However,*

in the results reported by the authors, the change rate of conductance drops as Δt becomes smaller. Can the authors explain this observation? Also, what were the parameters used for the experiments, such as the voltage amplitude of the pre- and post-neuron spikes? Why is the superimposition of the pulses necessary (which inevitably increases the power energy consumption)?

Answer 12: Thank you for raising this important topic. Figure RII-6a schematically shows the STDP procedure in a biological synapse^{9,18}. The relative timings (Δt) and the superimposition of neuronal spikes from the pre- and the post-synaptic neurons determine the weight change in the biological synapse, and Fig. RII-6b shows the STDP results from a real biological synapse¹⁹. It can be seen that the curvature is very similar to our emulating results in Fig. RII-6d and previous report⁹.

Fig. RII-6 | Schematic diagrams of STDP measurements. **a** Sketch of pre- and post-neurons connected by a synapse. The synaptic transmission is modulated by the delay time Δt of neuron spikes^{9,18}. **b** STDP results from a real biological synapse¹⁹. **c** Sketch of the STDP measurements for an FTJ memristor. **d** STDP measurements in the Ag/BTO/NSTO FTJ. Modulation of the FTJ conductance (ΔG) as a function of the delay (Δt) between pre- and post-synaptic spikes. The insets show the waveforms produced by the superimposition of pre- and post-synaptic spikes.

Therefore, to achieve the STDP based on an FTJ memristor, two voltage waveforms are designed and applied to the top and bottom electrodes (Fig. RII-6c),

which are necessary to simulate real neurons spikes from the pre- and post- synaptic neurons (Fig. RII-6a). The waveforms are made up of rectangular voltage pulses (1.5 V in amplitude, 20, 60, 100 ns in duration) followed by smooth slopes of opposite polarity (peak value of -1.5 V, and duration of 80, 240, 400 ns). With Δt close to 0, the pre- and post- spikes reach the synaptic device almost simultaneously, and the voltage drop on the FTJ will be close to 0. Thus, no conductance change is expected at $\Delta t = 0$, and this is the reason why the change rate of conductance drops as Δt becomes smaller.

References

1. Müller, J., Böske, T., Bräuhäus, D., Schröder, U., Böttger, U., Sundqvist, J., Kücher, P., Mikolajick, T. & Frey, L. Ferroelectric $Zr_{0.5}Hf_{0.5}O_2$ thin films for nonvolatile memory applications. *Appl. Phys. Lett.* **99**, 112901 (2011).
2. Ma, T. P. & Han, J.-P. Why is nonvolatile ferroelectric memory field-effect transistor still elusive? *IEEE Electron Device Lett.* **23**, 386-388 (2002).
3. Kim, D. J., Jo, J. Y., Kim, Y. S., Chang, Y. J., Lee, J. S., Yoon, J.-G., Song, T. K. & Noh, T. W. Polarization relaxation induced by a depolarization field in ultrathin ferroelectric $BaTiO_3$ capacitors. *Phys. Rev. Lett.* **95**, 237602 (2005).
4. Hadnagy, T. D. & Sheldon, D. J. Retention and Endurance effects of 4K and 64K FRAM Memories. *Integr. Ferroelectr.* **4**, 217-226 (1994).
5. Xi, Z. N., Jin, Q., Zheng, C. Y., Zhang, Y. C., Lu, C. J., Li, Q., Li, S. D., Dai, J. Y. & Wen, Z. High-temperature tunneling electroresistance in metal/ferroelectric/semiconductor tunnel junctions. *Appl. Phys. Lett.* **111**, 132905 (2017).
6. Choi, K. J., Biegalski, M., Li, Y. L., Sharan, A., Schubert, J., Uecker, R., Reiche, P., Chen, Y. B., Pan, X. Q., Gopalan, V., Chen, L. Q., Schlom, D. G. & Eom, C. B. Enhancement of ferroelectricity in strained $BaTiO_3$ thin films. *Science* **306**, 1005-1009 (2004).
7. Guo, R., Zhou, Y. X., Wu, L. J., Wang, Z. R., Lim, Z., Yan, X. B., Lin, W. N., Wang, H., Yoong, H. Y., Chen, S. H., Ariando, Venkatesan, T., Wang, J., Chow, G. M., Gruverman, A., Miao, X. S., Zhu, Y. M. & Chen, J. S. Control of synaptic plasticity learning of ferroelectric tunnel memristor by nanoscale interface engineering. *ACS Appl. Mater. Interfaces* **10**, 12862-12869 (2018).
8. Yang, Y. C., Pan, F., Liu, Q., Liu, M. & Zeng, F. Fully room-temperature-fabricated nonvolatile resistive memory for ultrafast and

- high-density memory application. *Nano Lett.* **9**, 1636-1643 (2009).
9. Boyn, S., Grollier, J., Lecerf, G., Xu, B., Locatelli, N., Fusil, S., Girod, S., Carretero, C., Garcia, K., Xavier, S., Tomas, J., Bellaiche, L., Bibes, M., Barthelemy, A., Saighi, S. & Garcia, V. Learning through ferroelectric domain dynamics in solid-state synapses. *Nat. Commun.* **8**, 14736 (2017).
 10. Jo, J. Y., Han, H. S., Yoon, J. G., Song, T. K., Kim, S. H. & Noh, T. W. Domain switching kinetics in disordered ferroelectric thin films. *Phys. Rev. Lett.* **99**, 267602 (2007).
 11. Boyn, S., Chanthbouala, A., Girod, S., Carrétéro, C., Barthélémy, A., Bibes, M., Grollier, J., Fusil, S. & Garcia, V. Real-time switching dynamics of ferroelectric tunnel junctions under single-shot voltage pulses. *Appl. Phys. Lett.* **113**, 232902 (2018).
 12. Rao, F., Ding, K. Y., Zhou, Y. X., Zheng, Y. H., Xia, M. J., Lv, S. L., Song, Z. T., Feng, S. L., Ronneberger, I., Mazzarello, R., Zhang, W. & Ma, E. Reducing the stochasticity of crystal nucleation to enable subnanosecond memory writing. *Science* **358**, 1423-1427 (2017).
 13. Wang, C., Wu, H. Q., Gao, B., Wu, W., Dai, L. J., Li, X. Y. & Qian, H. Ultrafast RESET Analysis of HfOx-Based RRAM by Sub-Nanosecond Pulses. *Adv. Electron. Mater.* **3**, 1700263 (2017).
 14. Åkerman, J. Toward a universal memory. *Science* **308**, 508-510 (2005).
 15. Ielmini, D. & Wong, H.-S. P. In-memory computing with resistive switching devices. *Nat. Electron.* **1**, 333-343 (2018).
 16. Wulf, W. A. & McKee, S. A. Hitting the memory wall: implications of the obvious. *ACM SIGARCH Computer Architecture News* **23**, 20-24 (1995).
 17. Zhang, B., Fan, F., Xue, W. H., Liu, G., Fu, Y., Zhuang, X. D., Xu, X. H., Gu, J. W., Li, R. W. & Chen, Y. Redox gated polymer memristive processing memory unit. *Nat. Commun.* **10**, 736 (2019).
 18. Kuzum, D., Yu, S. M. & Wong, H.-S. P. Synaptic electronics: materials, devices and applications. *Nanotechnology* **24**, 382001 (2013).
 19. Dan, Y. & Poo, M. M. Spike timing-dependent plasticity of neural circuits. *Neuron* **44**, 23-30 (2004).

Responses to Reviewer #3

Thank you very much for your pertinent comments and suggestions on our manuscript. We have carefully revised the manuscript and our responses to your comments are listed as follows:

Comment 1. *The ferroelectric layers were deposited by pulsed laser deposition on Nb-doped SrTiO₃ substrates. Transmission electron microscopy indicates that the BaTiO₃ layer is epitaxially grown on Nb:SrTiO₃ with a c/a ratio of 1.05 and Ti displacements suggest a ferroelectric polarization pointing toward the substrate. Regarding the piezoresponse force microscopy (PFM) experiments, the local PFM loop in Figure 1c shows a linear amplitude with voltage which is not consistent with a ferroelectric character. Instead the amplitude should drop at the coercive voltage and saturate for both polarization directions. I suggest to remove this Figure from the manuscript.*

Answer 1: Thank you for raising this important question. There are field-on and field-off modes in PFM measurements^{1,2}. Figure 1c of our previous manuscript displays a field-on mode PFM loop showing a linear amplitude with voltage, which is similar to the earlier reports^{3,4}.

Fig. RIII-1 | Ferroelectric characterizations by the PFM. PFM hysteresis loops. The orange and blue curves represent the PFM phase and the PFM amplitude, respectively.

To verify the ferroelectric property more accurately, we carried out PFM measurements in field-off mode, as shown in Fig. RIII-1. It is obvious that the amplitude drops at the coercive voltage and saturates for both polarization directions, consistent with a ferroelectric character as well as the results by the same

measurement mode⁵. Therefore, we replaced the results of the field-on mode by the new results of the field-off mode.

Figure RIII-1 is added as **Fig. 1g** in the revised manuscript.

Comment 2. *In the PFM images in Figure 1d and 1e, the authors compare the signal from domains written with positive and negative voltages. They show a clear 180-degree phase contrast between domains of opposite orientation and a similar PFM amplitude for both directions. I would suggest to show the PFM signal (phase and amplitude) in the virgin state as well to corroborate the transmission electron microscopy observations, i.e. downward polarization of BaTiO₃.*

Answer 2: Following your advice, the PFM phase and amplitude images within both the virgin region and the patterned region are measured, as shown in Fig. RIII-2. The domain structure of the $3 \times 3 \mu\text{m}^2$ area is patterned by -6 V and then the central $1 \times 1 \mu\text{m}^2$ with +6 V. The outer region is still in virgin state. It is clear that the virgin state of BTO shows a downward polarization, which is consistent with the STEM results.

Figure RIII-2b, c are added as **Fig. 1h, i**. The corresponding discussions were added in **Lines 116-117 of Page 5** of the revised manuscript.

Fig. RIII-2 | Ferroelectric characterizations by the PFM. **a** Protocol for domain patterning. **b** PFM phase and **c** PFM amplitude images recorded after writing an area of $3 \times 3 \mu\text{m}^2$ with -6 V and then the central $1 \times 1 \mu\text{m}^2$ with +6 V.

Comment 3. *In the geometry of the pulse-voltage experiments conducted by Ma et al., it is not clear if short voltage pulses are well transmitted across the tunnel junction or if their shape is modified when reaching the tunnel barrier. In order to make sure that the applied voltage pulse with a duration of 540ps (Figure S1a) is transmitted to the junction, the authors could conduct real-time current measurements during the application of the pulse (see Boyn et al., Appl. Phys. Lett. 113, 232902 (2018)). This would enable to measure the actual current transmitted during the voltage pulses to*

extract the write current densities. They should at least make a comment about it in the manuscript.

Answer 3: Thank you for raising the key technical issue. Following your suggestion, we conducted a real-time electrical measurement setup similar to that in the literature⁶⁻⁸, as shown in Fig. RIII-3a. A pulse generator (Tektronix PSPL10300B with the shortest pulse duration about 540 ps) delivers voltage pulses with different amplitudes and durations to induce resistance switchings in the FTJs. A Keithley 2410 SourceMeter was used to monitor the resistance change of the FTJs after applying write voltage pulses. An oscilloscope (Tektronix DSA70804 with a bandwidth of 8 GHz) was utilized to verify the waveforms applied to the FTJs. A DC/RF switch (Radiall's RAMSES SPDT switch, 0-18 GHz) was used to separate the DC and RF circuit signals. To protect the oscilloscope against overvoltage, -10 dB and -6 dB attenuators are connected before Channel 1 and Channel 2, respectively.

In this way, we can record the voltage pulse applied to the FTJ top electrode using Channel 1 of the oscilloscope. The signal transmitted through the FTJ is also recorded by the oscilloscope Channel 2. For example, Fig. RIII-3b shows that a voltage pulse of 540 ps in duration and 15 V in amplitude was successfully applied to the FTJ. While the signal transmitted through the FTJ shows a pulse duration of about 600 ps, as shown in Fig. RIII-3c. In other words, the RC delay τ_{RC} extends the 540 ps pulse to 600 ps, and τ_{RC} is estimated to be about $(600-540)/2 = 30$ ps. Compared with the sub-nanosecond pulse signal, such a small RC delay would not affect the conclusions obviously.

According to the results mentioned above, we changed the sub-nanosecond resistive switching time from 540 ps to 600 ps in the revised manuscript.

In addition, based on Fig. RIII-3c, the write current density can be estimated as $J \approx 4 \times 10^3$ A/cm². Figure RIII-3 is added as **Supplementary Fig. S3**. The experimental results and relevant descriptions have been added in **Lines 392-402 of Page 19** of the **Methods** section of the revised manuscript and the **S3** of the revised Supplementary information.

Fig. RIII-3 | Real-time electrical measurements. **a** Schematic description of the real-time electrical measurement setup. **b** Voltage pulse of 540 ps in duration and 15 V in amplitude applied to the FTJ top electrode. **c** Signal transmitted through the FTJ with a duration of 600 ps.

Comment 4. Page 8, line 152. *How did the authors estimate their write current densities of $4 \times 10^3 \text{ Acm}^{-2}$ during the application of the 540-ps voltage pulse? See my previous point.*

Answer 4: Here, as discussed in detail in our response to your Comment 3, a real-time current measurement was carried out. The write current density is about $4 \times 10^3 \text{ A/cm}^2$. Figure RIII-3 is added as **Supplementary Fig. S3**. The experimental results and relevant descriptions have been added in **S3** of the revised Supplementary information.

Comment 5. Page 9, line 196. *“In FTJs, the ferroelectric switching has been successfully described by a modified Kolmogorov-Avrami-Ishibashi (KAI) model (14,33)”. When mentioning “modified KAI”, I believe that the authors are refereeing to the “Nucleation-limited-switching (NLS)” model. Indeed, the Lorentzian time distribution they extract in Figure 3c are the nucleation times for each electric field. This should be corrected. In addition, given the strong similarity with the paper from*

Boyn et al., Nature Comm. (2017) (Ref. 26), the authors should refer to this paper when presenting this model and results.

Answer 5: We appreciate the reviewer's careful review very much. The "modified KAI model" mentioned in our previous manuscript should be the "nucleation-limited-switching (NLS) model". The correction has been made to the revised version. Following your advice, we refer to the paper (Boyn et al., Nature Comm. (2017) (Ref. 27)) when presenting this model and result.

The relevant descriptions have been added in **Lines 220, 231 of Page 11** and **Line 404 of Page 19** of the revised manuscript.

Comment 6. *The caption of Figure 5d is wrong and must be corrected as it is not the inverse of the coercive voltage vs. pulse duration but the negative coercive voltage vs. pulse duration.*

Answer 6: Thank you for pointing out this mistake. The **caption of Fig. 5d** was corrected by using the absolute value of coercive voltage $|V_c^-|$ versus the pulse duration t_d .

Comment 7. *The details of the growth process of Ag and definition of 70-micron pads are missing.*

Answer 7: Following your advice, the details of the growth of Ag were added. The Ag top electrodes of ~ 70 μm in diameter and ~ 30 nm in thickness were sputtered on the BTO/NSTO heterostructures through a shadow mask.

The related descriptions were added in **Line 381 of Page 18 to Line 383 of Page 19 of Methods** section of the revised manuscript.

Comment 8. *The symbol for Celsius degrees does not appear in the pdf.*

Answer 8: The missing symbol for Celsius degrees was caused by the conversion of the word file into the PDF format. We have corrected it.

References

1. Vasudevan, R. K., Balke, N., Maksymovych, P., Jesse, S. & Kalinin, S. V. Ferroelectric or non-ferroelectric: Why so many materials exhibit "ferroelectricity" on the nanoscale. *Appl. Phys. Rev.* **4**, 021302 (2017).
2. Strelcov, E., Kim, Y., Yang, J. C., Chu, Y. H., Yu, P., Lu, X., Jesse, S. & Kalinin, S. V. Role of measurement voltage on hysteresis loop shape in piezoresponse force

- microscopy. *Appl. Phys. Lett.* **101**, 192902 (2012).
3. Chen, B., Zheng, X. J., Yang, M. J., Zhou, Y., Kundu, S., Shi, J., Zhu, K. & Priya, S. Interface band structure engineering by ferroelectric polarization in perovskite solar cells. *Nano Energy* **13**, 582-591 (2015).
 4. Kundu, S., Clavel, M., Biswas, P., Chen, B., Song, H. C., Kumar, P., Halder, N. N., Hudait, M. K., Banerji, P., Sanghadasa, M. & Priya, S. Lead-free epitaxial ferroelectric material integration on semiconducting (100) Nb-doped SrTiO₃ for low-power non-volatile memory and efficient ultraviolet ray detection. *Sci. Rep.* **5**, 12415 (2015).
 5. Wen, Z., Li, C., Wu, D., Li, A. D. & Ming, N. B. Ferroelectric-field-effect-enhanced electroresistance in metal/ferroelectric/semiconductor tunnel junctions. *Nat. Mater.* **12**, 617 (2013).
 6. Boyn, S., Chanthbouala, A., Girod, S., Carrétéro, C., Barthélémy, A., Bibes, M., Grollier, J., Fusil, S. & Garcia, V. Real-time switching dynamics of ferroelectric tunnel junctions under single-shot voltage pulses. *Appl. Phys. Lett.* **113**, 232902 (2018).
 7. Rao, F., Ding, K. Y., Zhou, Y. X., Zheng, Y. H., Xia, M. J., Lv, S. L., Song, Z. T., Feng, S. L., Ronneberger, I., Mazzarello, R., Zhang, W. & Ma, E. Reducing the stochasticity of crystal nucleation to enable subnanosecond memory writing. *Science* **358**, 1423-1427 (2017).
 8. Wang, C., Wu, H. Q., Gao, B., Wu, W., Dai, L. J., Li, X. Y. & Qian, H. Ultrafast RESET Analysis of HfO_x-Based RRAM by Sub-Nanosecond Pulses. *Adv. Electron. Mater.* **3**, 1700263 (2017).

Reviewers' comments:

Reviewer #1 (Remarks to the Author):

The authors have reasonably addressed my major comments and other reviewers' comments and particularly have taken into account the possible reasons of the sub-nanosecond switching as well as the limit of the measurement system. The manuscript has been revised appropriately. Thus, the reviewer agrees that the manuscript can be now published in Nature Communications as it is.

Reviewer #2 (Remarks to the Author):

The authors have revised the manuscripts according to the comments from the reviewers. The quality of the manuscript has been improved, but there are remaining issues that should be addressed.

1. The concept of 'universal memory'. The authors claimed that since their device possesses the best attributes of all different commercial memories (static RAM, dynamic RAM, and flash) — high speed, low energy consumption, high endurance, high density, non-volatility, etc. However, the endurance data is far from the requirements for a usable memory ($1E12$ to $1E16$ cycles). Even if the properties of an individual device is great, it is not demonstrated if the device performance is uniform from device to device and from cycle to cycle. The variability is among the most important parameter that should be demonstrated using arrays in order to make a memory out of this device
2. The authors claimed that the device can be used for analog computing, merely based on the fact that the device shows analog tuning behavior (which, by the way, has been demonstrated routinely by others in this community). However, how stable are these multiple resistance states? How the IV relationship will be used for computing and what are other requirements on the device properties? These questions should be answered with solid experimental data.
3. The 'STDP' curve shown in Fig. 4b is not strict STDP behavior, the increase or decrease of the weight should be monotonous with the change of the time interval.
4. It is not clear what the test structure is for the speed test. In other words, it is still not clear if the sub-ns has been delivered to the junction area.

Reviewer #3 (Remarks to the Author):

I am completely satisfied with the response of the authors to the points raised by the three Referees. I am very impressed by the quality of this experimental work and fully recommend its publication in Nature Communications.

A list of changes

1. In **Lines 2-3 of Page 1** of the revised manuscript, we all agree to change the author “Zhen Luo” to be the second author and add the authors “Letian Zhao” and “Xi Jin” because of their contributions on the artificial neuron network simulations.
2. In **Line 19 and Lines 33-34 of Page 2**, the statements about “universal memory” were removed.
3. In **Lines 78-82 of Page 4**, the sentence “Based on the experimental performances..... through an online supervised learning.” was added in the revised manuscript.
4. In **Lines 172-175 of Page 10**, the sentence “In addition, the FTJs also..... shown in Supplementary Fig. S5c.” was added.
5. In **Lines 326-335 of Page 18**, the sentences “Artificial neural network simulation with..... future neuromorphic networks (Supplementary Fig. S15).” were added.
6. In **Lines 342-343 of Page 19**, the sentence “and the ANN simulation a high recognition accuracy of > 90 % on MNIST digits.” was added.
7. In **Line 377 of Page 20 to Line 385 of Page 21**, the sentences “**Neural network simulations.....** for the weight update (Supplementary Fig. S14).” were added.
8. In the Supplementary information, **S5 “Endurance and device-to-device variability”** and **S12 “Artificial neural network (ANN) simulation with FTJs”** were added.

There are also some minor revisions about the sentences to improve English language as well as several reference updates, which are not listed here.

Responses to Reviewer #2

Thank you very much for your valuable comments on our manuscript. According to your comments, several important experimental and simulation investigations have been carried out. We have prepared a point-by-point response to your comments and carefully revised the manuscript accordingly, including:

1) **Memory**: The endurance and device variability were experimentally characterized systematically. We obtained repeatable resistance switchings up to 10^8 - 10^9 cycles, one of the best endurance results among the reported FTJs. Although the current endurance is not sufficient for universal memories, it is higher than that of the flash memories and would meet the desired endurance metric of artificial synapses for analog computings¹. Small resistance fluctuations from device to device (relative standard deviation RSD $\sim 5\%$) and from cycle to cycle (RSD $\sim 2\%$) demonstrate the good reproducibility and uniformity of the FTJs.

2) **Computing**: The evolution of the conductance with voltage pulses, I - V curves at different conductance states, and the device variations, which are important for the analog computing, have been characterized experimentally. Then, based on these realistic device properties, an artificial neural network was simulated to perform an online supervised learning on the Modified National Institute of Standard and Technology handwritten digits. A high recognition accuracy ($> 90\%$) can be obtained based on above obtained experimental performances of our FTJs, highlighting their possible potential for constructing neuromorphic networks.

3) **STDP**: A systematic literature review on both biological and electrical STDP behaviors has been carried out. It turns out that there are mainly two types of STDP curvatures, monotonous and non-monotonous weight variations, with the change of the time interval Δt ($\Delta t \geq 0$ or $\Delta t \leq 0$).

4) **Real-time electrical measurement**: To give a clearer description on how we set up the test structure, we summarized some representative test structures utilized in literatures, which were used to check whether the sub-nanosecond pulses were delivered successfully to the devices.

We wish that the revised version and the responses would be satisfactory to you.

Comment 1. *The concept of ‘universal memory’. The authors claimed that since their device possesses the best attributes of all different commercial memories (static RAM, dynamic RAM, and flash) — high speed, low energy consumption, high endurance, high density, non-volatility, etc. However, the endurance data is far from the requirements for a usable memory ($1E12$ to $1E16$*

cycles). Even if the properties of an individual device is great, it is not demonstrated if the device performance is uniform from device to device and from cycle to cycle. The variability is among the most important parameter that should be demonstrated using arrays in order to make a memory out of this device.

Answer 1: We thank the reviewer for raising the important topics of endurance and variability for memory applications. For our FTJs, the endurance up to 10^8 - 10^9 cycles were observed, one of the best endurance results among the reported FTJs²⁻⁴. Although the current endurance is not enough for a universal memory, it is much better than flash memories and would be satisfied for neuromorphic computing^{1,5}. In addition, small resistance fluctuations from device to device (relative standard deviation RSD $\sim 5\%$) and from cycle to cycle (RSD $\sim 2\%$) demonstrate the good reproducibility and uniformity of the FTJs.

The details are as follows:

1) Endurance. As the reviewer pointed out, a high endurance $>10^{12}$ is required for a universal memory. While for ferroelectric-based memories, it is noted that the ferroelectric fatigue had been an issue affecting their endurance. Fortunately, after extensive studies by many researchers, the endurance in ferroelectric films with several hundred nanometers thick can usually reach 10^{12} to 10^{14} or even higher, such as the results reported in BaTiO_3 ⁶, $\text{Pb}(\text{Zr}, \text{Ti})\text{O}_3$ ⁷, and $\text{SrBi}_2\text{Ta}_2\text{O}_9$ ⁸. On the other hand, for the recently developed FTJs using ferroelectric barriers as thin as several nanometers, the endurance is currently around 10^5 - 10^9 cycles²⁻⁴. To test the endurance limit of our FTJs, a function generator (Agilent 33220A) was utilized to generate square waveforms ($\pm 3\text{V}$, 100 ns) to flip the ferroelectric polarization repeatedly, and the resistances were recorded after the voltage pulses. It was observed that the FTJs show repeatable resistance switchings up to 10^8 - 10^9 cycles, and the representative results from two different FTJs are shown in Fig. R1. In particular, it is worth mentioning that the corresponding endurance results in FTJs ($\sim 10^5$ - 10^9) are similar to those of the thick ferroelectric films in the early research stage. Namely, higher endurance $>10^{12}$ may be expected after systematically dedicated investigations. We will pay more attention to investigate the endurance of FTJs in near future. Thanks to the reviewer for raising this important topic.

At present, the endurance performance $\sim 10^8$ - 10^9 of FTJs is already higher than that of the flash memories (about 10^5 orders of magnitude)⁵, and it would be also sufficient to meet the desired endurance metric $\sim 10^9$ of artificial synapses in neural networks for analog computings¹. For

example, it has been pointed out that an endurance of 10^5 may be required to train neural networks in an online fashion^{1,9}, while the device endurance is less of a concern in offline learning, since the synaptic weights do not need to be frequently updated¹. However, further dedicated researches on endurance are necessary to make FTJs usable as a universal memory. Therefore, the statements of “universal memory” in the previous version were deleted. In addition, the endurance results in Fig. R1 are added as **Supplementary Fig. S5a, b**. The corresponding discussions were added in **Lines 172-175 of Page 10** of the revised manuscript.

Fig. R1 | Endurance measurements. a, b Reproducible resistance switchings up to 10^8 - 10^9 of two representative FTJs by cycling pulse voltages ($\pm 3V$, 100 ns) using a function generator (Agilent 33220A).

2) Variability. As the reviewer pointed out, uniform device performances from device to device and from cycle to cycle are critical for constructing reliable memory arrays. For our FTJs, the cycle-to-cycle variability can be experimentally estimated from the repeatable resistance switchings among different resistance states shown in Fig. 2b, d in main text and Fig. S2d, f in supplementary information. For example, the resistances and the relative standard deviations (RSD) for the five different resistance states obtained from Fig. 2b are listed in Table R1. After analyzing all multi-state resistance switching results, a RSD of $\sim 2\%$ was obtained for the multi-state resistance switchings, demonstrating the good cycle-to-cycle uniformity.

However, on the other hand, building a real memory array for evaluating the device-to-device variability is far beyond our technical capability at present. We are only able to fabricate many individual FTJs and test their performances one by one. Fig. R2 shows the resistance switching results of 20 different Ag/BTO/NSTO FTJ devices. Despite of the subtle differences among the

manually controlled growth conditions for different batches of FTJs as well as the large size (~ 70 μm in diameter) of the devices, the resistance fluctuations and RSD of ON ($4.08 \times 10^4 \Omega - 4.71 \times 10^4 \Omega$, RSD $\sim 4.3\%$) and OFF ($4.80 \times 10^6 \Omega - 5.71 \times 10^6 \Omega$, RSD $\sim 5.5\%$) states are small for the FTJ samples.

Table R1. Resistances and relative standard deviations (RSD) for the five different resistance states from Fig. 2b in main text.

State	Resistance	RSD
1	$6.37 \times 10^6 - 6.98 \times 10^6$	2.4%
2	$2.21 \times 10^6 - 2.28 \times 10^6$	0.8%
3	$6.46 \times 10^5 - 6.67 \times 10^5$	0.8%
4	$2.08 \times 10^5 - 2.25 \times 10^5$	1.8%
5	$4.35 \times 10^4 - 4.66 \times 10^4$	2.2%

Fig. R2 | Device-to-device variability. Resistances of ON and OFF states for 20 different FTJ devices.

The good uniformity in FTJs should be related to the intrinsic ferroelectric nature induced resistance switchings. In the future, much better control on the uniformity can be expected during

the industrial semiconductor manufacturing by standardizing the fabrication conditions and reducing the device sizes.

In addition, it is worth mentioning that the analog computing using memristor arrays can bear certain variabilities from device to device and from cycle to cycle, as indicated by earlier theoretical and experimental reports on memristor crossbars^{10,11}. For our FTJs, the variability performances would be useful for the analog computing to achieve a high recognition accuracy (> 90 %) according to the theoretical simulations of an artificial neural network (ANN) (see details in our response to Comment 2).

Figure **R2** is added as **Supplementary Fig. S5c**. The corresponding discussions were added in **Lines 172-175 of Page 10** of the revised manuscript and **S5** of the revised Supplementary Information, respectively.

Comment 2. *The authors claimed that the device can be used for analog computing, merely based on the fact that the device shows analog tuning behavior (which, by the way, has been demonstrated routinely by others in this community). However, how stable are these multiple resistance states? How the $I-V$ relationship will be used for computing and what are other requirements on the device properties? These questions should be answered with solid experimental data.*

Answer 2: We thank the reviewer for raising the important questions on the analog computing. Building a memristor array with stable multi-state switching, good uniformity, *etc.* is the ideal goal of the field of analog computing and thus a great challenge for the future. Therefore, the evolution of the conductance with voltage pulses, $I-V$ curves at different states and the device variations, which are important for the analog computing, have been characterized experimentally. However, owing to our limited technical capability at present, we are only able to demonstrate the promising potential of FTJs for applications in the analog computing by carrying on an artificial neural network (ANN) simulation based on the above realistic performances of the individual FTJ. We hope that our following answers would be satisfactory to the reviewer, and further in-depth investigations will be carried out following the reviewer's constructive suggestions.

Here, the artificial neural network was simulated to perform an online supervised learning on the Modified National Institute of Standard and Technology (MNIST) database. Especially, the realistic device properties including the $I-V$ relationship, the evolution of the conductance with

voltage pulses, and the experimental device variations were used to build the device behavioral model for the ANN simulation. Interestingly, a high recognition accuracy ($> 90\%$) can be obtained based on the experimental performances of our FTJs (64 states, ON/OFF ratio ~ 80 , cycle-to-cycle standard deviation relative to the entire conductance range $\Delta \sim 2.7\%$, device-to-device variation RSD $\sim 5\%$), demonstrating their potential ability in constructing neuromorphic computing networks. The details are described as follows.

1) ANN Simulation. Similar to earlier reports^{10,11}, a two-layer perceptron with 784 input neurons, 100 hidden neurons, and 10 output neurons was simulated to implement the online supervised learning on the MNIST handwritten digits database, as shown in Fig. R3. The 784 input neurons correspond to a 28×28 MNIST image, and the 10 output neurons correspond to 10 classes of digits ($c = 0 - 9$). Generally, for a real memristor crossbar, the inference or classification of a MNIST image was performed by biasing the top electrode of memristors in the first layer with a set of input voltages (V_{input} , corresponding to small reading voltages without affecting memristor states) whose amplitudes encode an image, then reading the currents from the bottom electrodes of devices in the final layer.

Fig. R3 | Artificial neural network. Schematic diagram of a two-layer neural network.

In terms of stochastic gradient descent (SGD) and back propagation algorithms¹⁰, the flow chart of the training is schematically illustrated in Fig. R4. The training is composed of two stages: feedforward inference and feedback weight (W) update, and the weight update is carried out by software but based on the realistic device performances. For each training cycle, 128 images

randomly selected from 60000 MNIST digits are set as a batch (B), and the indicator n is the number (1-128) of image in each batch. The multilayer inference is performed layer by layer sequentially. The input voltage vector (X) to the first layer is a feature vector from the MNIST dataset. For example, a pixel of a grayscale 0 – 255 corresponds to an input voltage 0 – 0.05 V. The input vector for the subsequent layer is obtained based on the output vector of the first layer (I).

Fig. R4 | Training algorithm. Flow chart of the training process.

For the two-layer perceptron, the hidden input $X^2(n)$ and output $F_c(n)$ are obtained using Equations 1-3^{10,12}.

$$I_j^l = \sum_i W_{ij}^l X_i^l . \quad (1)$$

$$X_j^2 = a[\text{ReLU} (I_j^1)]. \quad (2)$$

$$F_c(n) = \text{softmax}(\sum_{j=1}^{100} W_{cj}^2 X_j^2) . \quad (3)$$

Here, a is a scaling factor to match the output of the first layer to the input of the second layer¹⁰. W^1 and W^2 are weight matrices of the first and second layers, respectively. The ReLU function in Equation 2, as the activation function, is defined as

$$\text{ReLU}(x) = \max(0, x) . \quad (4)$$

The softmax function in Equation 3 is defined as:

$$y_c(n) = \frac{e^{mI_c(n)}}{\sum_{p=1}^{10} e^{mI_p(n)}} , \quad (5)$$

where $y_c(n)$ is the probability that Image n belongs to Class c . Then, the cross-entropy loss function $\xi(n)$ is calculated by using Equation 6 to evaluate the difference between the calculated output $F_c(n)$ and the ideal result possibility $t_c(n)$.

$$\xi(T(n), F(n)) = -\sum_{c=1}^{10} t_c(n) \log[F_c(n)]. \quad (6)$$

The desired weight updates (ΔW) of the synaptic device in each layer are calculated by using Equations 7 and 8.

$$\delta_j^l = \begin{cases} \frac{\partial \xi}{\partial I_j^l}, & l=2 \\ \frac{\partial \sigma}{\partial I_j^l} \sum_i W_{ij}^l \delta_i^{l+1}, & l=1 \end{cases} \quad (7)$$

$$\Delta W^l = \eta \sum_{n=1}^B \frac{\delta^l(n) X^l(n)^T}{B}. \quad (8)$$

Here, σ is the nonlinear activation function of the hidden layer and η is the learning rate obtained from the back propagating algorithm.

2) Device behavioral model and multi-state stability. To implement the SGD algorithm in the memristor crossbar in which the synaptic weight could be positive and negative, the synaptic weight can be encoded as the conductance difference between two memristors¹⁰ by Equation 9.

$$W_{ij}^l = G_{ij}^{l+} - G_{ij}^{l-}. \quad (9)$$

Namely, each synapse is implemented with two memristors, so that the total number of memristors in the cross bar is $(784 \times 100 + 100 \times 10) \times 2 = 158800$. Therefore, the realistic device performances, including conductance vs. pulse number, I - V relationship, device-to-device variability, cycle-to-cycle variation, *etc.*, will affect how the accuracy of the ΔW can be applied to the synapse devices of the ANN. Typically, the weight update of each memristor can be achieved by applying voltage pulses. Here, based on the I - V relationship and the R - V_{pulse} curve, we set a range of conductance switching for computing simulation by applying voltage pulses with incremental amplitude (from -0.3 V to -1.8 V with a step of 30 mV, -1.8 V to -3.1 V with a step of 100 mV), as shown in Fig. R5a in which the device shows 64 conductance states (read at 0.05 V). The measurements were repeated by 20 times, and a cycle-to-cycle variation of $\Delta \sim 2.7\%$ was obtained, which demonstrates the good stabilities for different states.

Fig. R5 | Device behavioral model and multi-state stability. **a.** Conductance vs. pulse number measured for 20 times. **b.** Average conductance vs. pulse number.

Fig. R6 | Simulated pattern recognition accuracy of the two-layer ANN. Blue curve: Simulations based on the FTJ device behavioral model memristors. Orange curve: Simulations based on ideal synaptic devices.

Accordingly, the average conductance vs. pulse number can be obtained, as shown in Fig. R5b, which was utilized as the device behavioral model for simulations. The cycle-to-cycle and device-to-device variations can be included as random fluctuations following Gaussian distribution in certain ranges. Fig. R6 shows the simulation results for the ANN based on the realistic device behavioral model. By comparison, the simulation for an ANN using the ideal synapse (*i.e.*, a memristor which could be tuned to any resistance state without device variation) was also carried out. It can be seen that the ANN based on the FTJs (with a device-to-device variation RSD $\sim 5\%$ estimated from experimental results in Fig. R2) shows a pattern recognition accuracy $> 90\%$, which is close to the recognition accuracy $\sim 97\%$ calculated on the basis of an ideal ANN.

It can be seen from Fig. R6 that the analog computing using memristor arrays can bear certain device variabilities. The effects of the cycle-to-cycle and the device-to-device variations on recognition accuracies have also been simulated systematically, as shown in Fig. R7. Fig. R7a shows the recognition accuracy vs. cycle-to-cycle variation with a fixed device-to-device variation (RSD $\sim 5\%$), and Fig. R7b shows the recognition accuracy vs. device-to-device variation with the experimental cycle-to-cycle variation from Fig. R5. It can be seen that the recognition accuracy decreases with increasing cycle-to-cycle and device-to-device variations. Based on the realistic device behavioral model in Fig. R5, it is still $\sim 87\%$ even supposing the device-to-device variation RSD to be $\sim 35\%$, consistent with the earlier report¹³.

Fig. R7 | Effect of device variability on simulated recognition accuracy. **a.** Recognition accuracy vs. cycle-to-cycle variation with a fixed device-to-device variation (RSD $\sim 5\%$). **b.** Recognition accuracy vs. device-to-device variation with the experimental cycle-to-cycle variation from Fig. R5.

Fig. R8 | $I-V$ curves of the Ag/BTO/NSTO FTJ at different states. **a.** $I-V$ curves in between -0.1 V and 0.1 V. **b.** $I-V$ curves in between 0 V and 0.05 V.

3) I - V relationship. As shown in Equation 1, the forward inference in the computing simulation depends on the relationship between input voltage and output current in the input voltage (V_{input}) range (*i.e.*, the I - V relationship of the memristor at low biases). In other words, the I - V relationship will affect the recognition accuracy by affecting the forward inference. Up to date, there is no report using nonlinear I - V for ANN simulations, and the ANN simulations mentioned above are also performed by assuming a linear I - V relationship in the V_{input} range from 0 V to a small positive bias voltage. Here, to evaluate the effect of realistic I - V relationship on computing, the representative I - V curves of different resistance states were experimentally characterized, as shown in Fig. R8. It can be seen that the nonlinearity decreases with decreasing bias voltage (see Fig. R8a), and it becomes very linear between a V_{input} range from 0 V to 0.05 V (see Fig. R8b). In other words, the nonlinearity decreases with decreasing V_{input} range. Therefore, we tried to use the realistic I - V curves at different V_{input} ranges for ANN simulations to evaluate how the I - V nonlinearity affects the computing. With the experimental cycle-to-cycle variation ($\Delta \sim 2.7\%$) obtained from Fig. R5 and the device-to-device variation estimated from Fig. R2 (RSD $\sim 5\%$), the recognition accuracy vs. input voltage range is simulated, as displayed in Fig. R9. It can be seen that the recognition accuracy increases with decreasing input voltage range (*i.e.*, with increasing I - V linearity), and it is $> 90\%$ below 0.05 V.

Fig. R9 | Simulated recognition accuracy with different input voltage ranges.

For comparison, previous ANN simulations based on realistic performances of different memristors can reach a recognition accuracy $\sim 10\% - 91\%$ on MNIST digits^{11,14}. Thus, one can see that our FTJ could be a good analog synaptic device for neuromorphic hardware systems. Figs. **R3**, **R5**, **R6**, and **R7b** are added as **Supplementary Figs. S13-S15**. The corresponding discussions

were added in **Lines 78-82 of Page 4**, **Lines 326-335 of Page 18**, **Lines 342-343 of Page 19**, and **Line 377 of Page 20** to **Line 385 of Page 21** of the revised manuscript and **S12** of the revised Supplementary Information, respectively.

Comment 3. *The ‘STDP’ curve shown in Fig. 4b is not strict STDP behavior, the increase or decrease of the weight should be monotonous with the change of the time interval.*

Answer 3: Thank you for raising this important topic. To get a more comprehensive understanding of STDP, we have carried out a systematic literature review on both biological and electrical STDP behaviors. It turns out that there are mainly two types of STDP curvatures, monotonous and non-monotonous weight variations, with the change of the time interval Δt ($\Delta t \geq 0$ or $\Delta t \leq 0$). They are described in details as follows.

Fig. R10 | Representative STDP curves with monotonous curvatures for $\Delta t \geq 0$ or $\Delta t \leq 0$. **a, b** STDP curves experimentally tested in biological synapses from **a** rat hippocampal neurons¹⁵ and **b** frog tectal neurons¹⁶. **c, d** STDP of electrical synapse devices based on **c** *h*-BN/WSe₂ heterostructure¹⁷ and **d** ionic gated MoS₂¹⁸.

1) Monotonous STDP behavior. As pointed out by the reviewer, there are many STDP curves showing monotonous variation of the weight with the change of Δt ($\Delta t \geq 0$ or $\Delta t \leq 0$)^{17,18}, and some representative STDP curves tested in biological and electrical synapses are shown in Fig. R10a, b^{15,16} and Fig. R10c, d^{17,18}, respectively. It can be seen that the largest changes in synaptic

weight (strengthening or weakening) occur at small Δt , and there is a sharp transition from strengthening to weakening of synaptic weights as Δt passes through zero.

2) Non-monotonous STDP behavior. There are also some of STDP results presenting non-monotonous curvatures with varying Δt ($\Delta t \geq 0$ or $\Delta t \leq 0$)¹⁹⁻²¹, similar to the STDP curves observed in our FTJ samples (Fig. 4b of the main text). Fig. R11a, b and Fig. R11c, d show the representative non-monotonous STDP curves measured in biological^{22,23} and electrical^{24,25} synapses, respectively. Distinctly, the maximum variation in synaptic weight occurs at a finite non-zero Δt , and the synaptic weight variation passes through zero continuously at $\Delta t = 0$. This actually could be understood based on the process of STDP. At $\Delta t = 0$, neuronal spikes from the pre- and the post-synaptic neurons reach the synapse simultaneously, and the corresponding voltage potential drop across the synapse will be close to zero if the pre- and post- spikes are usually very similar to each other²⁶. Therefore, with the effects from the pre- and post- spikes cancel with each other at $\Delta t = 0$, there will be no synaptic weight change in this case.

Fig. R11 | Representative STDP curves with non-monotonous curvatures for $\Delta t \geq 0$ or $\Delta t \leq 0$.

a, b STDP curves experimentally tested in biological synapses from **a** mossy fiber–granule cell synapse in rat²² and **b** frog tectal neurons²³. **c, d** STDP measurements of electrical synapse devices based on **c** Al/Cu-pMSSQ/Al heterostructure²⁴ and **d** Ag/PEDOT:PSS/Ta heterostructure²⁵.

In fact, according to the recent review article by Park *et al.*²⁷, there may be mainly four types

of STDP widely emulated by artificial synapses, including the ones with monotonous and non-monotonous curvatures discussed above, as shown in Fig. R12 (Fig. 2f of the review article by Park *et al.*²⁷). Therefore, it may not lead to misunderstanding by using the term “STDP” in our manuscript.

Fig. R12 | Four types of STDP forms widely emulated by artificial synapses. The figure is Fig. 2f of the very recent review article by Park *et al.*²⁷.

Comment 4. *It is not clear what the test structure is for the speed test. In other words, it is still not clear if the sub-ns has been delivered to the junction area.*

Answer 4: We are sorry that the real-time test structure was not described clearly in our previous version. To give a clearer description on how we set up the test structure, we summarized some representative test structures utilized in literatures, which are used to check whether the sub-nanosecond pulses were delivered successfully to the devices, as shown in Fig. R13²⁸⁻³¹. As described in these references, all these real-time electrical measurement setups are very similar and commonly utilized an oscilloscope to verify the waveform applied to the device.

According to the setups in Fig. R13, we built almost a same real-time electrical measurement setup, as shown in Fig. R14a. A pulse generator (Tektronix PSPL10300B with the shortest pulse duration about 540 ps) delivers voltage pulses with different amplitudes and durations to induce resistance switchings in the FTJs. A Keithley 2410 SourceMeter was used to monitor the resistance

change of the FTJs after applying write voltage pulses. An oscilloscope (Tektronix DSA70804 with a bandwidth of 8 GHz) was utilized to verify the waveforms applied to the FTJs. A DC/RF switch (Radiall's RAMSES SPDT switch, 0-18 GHz) was used to separate the DC and RF circuit signals. To protect the oscilloscope against overvoltage, -10 dB and -6 dB attenuators are connected before Channel 1 and Channel 2, respectively.

Fig. R13 | Real-time electrical measurement setups from different research groups²⁸⁻³¹. The oscilloscope is utilized to verify the waveform applied to the device. **a** is from [Science 358 (2017): 1423-1427]. **b** is from [Advanced Electronic Materials 3.12 (2017): 1700263]. **c** is from [Applied Physics Letters 113 (2018): 232902]. **d** is from [2016 IEEE Silicon Nanoelectronics Workshop (pp. 82-83)].

In this way, we can record the voltage pulse applied to the FTJ top electrode using Channel 1 of the oscilloscope. The signal transmitted through the FTJ is also recorded by the oscilloscope Channel 2. For example, Fig. R14b shows that a voltage pulse of 540 ps in duration and 15 V in amplitude was successfully applied to the FTJ. While the signal transmitted through the FTJ shows a pulse duration of about 600 ps, as shown in Fig. R14c. In other words, the RC delay τ_{RC} extends the

540 ps pulse to 600 ps, and τ_{RC} is estimated to be about $(600-540)/2 = 30$ ps. Compared with the sub-nanosecond pulse signal, such a small RC delay would not affect the conclusions obviously.

Fig. R14 | Real-time electrical measurements. **a** Schematic description of the real-time electrical measurement setup. **b** Voltage pulse of 540 ps in duration and 15 V in amplitude applied to the FTJ top electrode. **c** Signal transmitted through the FTJ with a duration of 600 ps.

References

1. Zhang, T., Yang, K., Xu, X. Y., Cai, Y. M., Yang, Y. C. & Huang, R. Memristive Devices and Networks for Brain-Inspired Computing. *Phys. Status. Solidi-R* **13**, 1900029 (2019).
2. Hu, W. J., Wang, Z. H., Yu, W. L. & Wu, T. Optically controlled electroresistance and electrically controlled photovoltage in ferroelectric tunnel junctions. *Nat. Commun.* **7**, 10808 (2016).
3. Guo, R., Wang, Z., Zeng, S. W., Han, K., Huang, L., Schlom, D. G., Venkatesan, T. & Chen, J. S. Functional ferroelectric tunnel junctions on silicon. *Sci. Rep.* **5**, 12576 (2015).
4. Max, B., Hoffmann, M., Slesazeck, S. & Mikolajick, T. Direct Correlation of Ferroelectric Properties and Memory Characteristics in Ferroelectric Tunnel Junctions. *IEEE J. Electron.*

- Devi.* (2019).
5. Chanthbouala, A., Crassous, A., Garcia, V., Bouzehouane, K., Fusil, S., Moya, X., Allibe, J., Dlubak, B., Grollier, J., Xavier, S., Deranlot, C., Moshar, A., Proksch, R., Mathur, N. D., Manuel, B. & Barthélémy, A. Solid-state memories based on ferroelectric tunnel junctions. *Nat. Nanotechnol.* **7**, 101 (2012).
 6. Chang, L. H. & Anderson, W. A. Stability of BaTiO₃ thin films on Si. *Appl. Surf. Sci.* **92**, 52-56 (1996).
 7. Wang, Y. K., Tseng, T. Y. & Lin, P. Enhanced ferroelectric properties of Pb(Zr_{0.53}Ti_{0.47})O₃ thin films on SrRuO₃/Ru/SiO₂/Si substrates. *Appl. Phys. Lett.* **80**, 3790-3792 (2002).
 8. De Araujo, C.-P., Cuchiaro, J. D., McMillan, L. D., Scott, M. C. & Scott, J. F. Fatigue-free ferroelectric capacitors with platinum electrodes. *Nature* **374**, 627 (1995).
 9. Cai, Y., Lin, Y. J., Xia, L. X., Chen, X. M., Han, S., Wang, Y. & Yang, H. Z. in *Proceedings of the 55th Annual Design Automation Conference*. 107 (ACM).
 10. Li, C., Belkin, D., Li, Y. N., Yan, P., Hu, M., Ge, N., Jiang, H., Montgomery, E., Lin, P. & Wang, Z. R. Efficient and self-adaptive in-situ learning in multilayer memristor neural networks. *Nat. Commun.* **9**, 2385 (2018).
 11. Kim, M.-K. & Lee, J.-S. Ferroelectric Analog Synaptic Transistors. *Nano Lett.* **19**, 2044-2050 (2019).
 12. Zhang, Q. T., Wu, H. Q., Yao, P., Zhang, W. Q., Gao, B., Deng, N. & Qian, H. Sign backpropagation: An on-chip learning algorithm for analog RRAM neuromorphic computing systems. *Neural Networks* **108**, 217-223 (2018).
 13. Yu, S. M., Chen, P.-Y., Cao, Y., Xia, L. X., Wang, Y. & Wu, H. Q. Scaling-up resistive synaptic arrays for neuro-inspired architecture: Challenges and prospect. *International Electron Devices Meeting (IEDM)*, 17.13.11-17.13.14 (IEEE, Washington, CA, USA, 2015).
 14. Jerry, M., Chen, P.-Y., Zhang, J. C., Sharma, P., Ni, K., Yu, S. M. & Datta, S. M. Ferroelectric FET analog synapse for acceleration of deep neural network training. *International Electron Devices Meeting (IEDM)*, 6.2.1-6.2.4 (IEEE, San Francisco, CA, USA, 2017).
 15. Bi, G. Q. & Poo, M. m. Synaptic modifications in cultured hippocampal neurons: dependence on spike timing, synaptic strength, and postsynaptic cell type. *J. Neurosci.* **18**, 10464-10472 (1998).
 16. Zhang, L. I., Tao, H. W., Holt, C. E., Harris, W. A. & Poo, M. M. A critical window for

- cooperation and competition among developing retinotectal synapses. *Nature* **395**, 37 (1998).
17. Seo, S., Jo, S.-H., Kim, S., Shim, J., Oh, S., Kim, J.-H., Heo, K., Choi, J.-W., Choi, C. & Oh, S. Artificial optic-neural synapse for colored and color-mixed pattern recognition. *Nat. Commun.* **9**, 5106 (2018).
 18. John, R. A., Liu, F., Chien, N. A., Kulkarni, M. R., Zhu, C., Fu, Q., Basu, A., Liu, Z. & Mathews, N. Synergistic Gating of Electro-Iono-Photoactive 2D Chalcogenide Neuristors: Coexistence of Hebbian and Homeostatic Synaptic Metaplasticity. *Adv. Mater.* **30**, 1800220 (2018).
 19. Boyn, S., Grollier, J., Lecerf, G., Xu, B., Locatelli, N., Fusil, S., Girod, S., Carretero, C., Garcia, K., Xavier, S., Tomas, J., Bellaiche, L., Bibes, M., Barthelemy, A., Saighi, S. & Garcia, V. Learning through ferroelectric domain dynamics in solid-state synapses. *Nat. Commun.* **8**, 14736 (2017).
 20. Prezioso, M., Bayat, F. M., Hoskins, B., Likharev, K. & Strukov, D. Self-adaptive spike-time-dependent plasticity of metal-oxide memristors. *Sci. Rep.* **6**, 21331 (2016).
 21. Yu, F., Zhu, L. Q., Xiao, H., Gao, W. T. & Guo, Y. B. Restickable Oxide Neuromorphic Transistors with Spike-Timing-Dependent Plasticity and Pavlovian Associative Learning Activities. *Adv. Func. Mater.* **28**, 1804025 (2018).
 22. Sgritta, M., Locatelli, F., Soda, T., Prestori, F. & D'Angelo, E. U. Hebbian spike-timing dependent plasticity at the cerebellar input stage. *J. Neurosci.* **37**, 2809-2823 (2017).
 23. Bell, C. C., Han, V. Z., Sugawara, Y. & Grant, K. Synaptic plasticity in a cerebellum-like structure depends on temporal order. *Nature* **387**, 278 (1997).
 24. Wu, C., Kim, T. W., Choi, H. Y., Strukov, D. B. & Yang, J. J. Flexible three-dimensional artificial synapse networks with correlated learning and trainable memory capability. *Nat. Commun.* **8**, 752 (2017).
 25. Li, S. Z., Zeng, F., Chen, C., Liu, H. Y., Tang, G. S., Gao, S., Song, C., Lin, Y. S., Pan, F. & Guo, D. Synaptic plasticity and learning behaviours mimicked through Ag interface movement in an Ag/conducting polymer/Ta memristive system. *J. Mater. Chem. C* **1**, 5292-5298 (2013).
 26. Kuzum, D., Yu, S. M. & Wong, H.-S. P. Synaptic electronics: materials, devices and applications. *Nanotechnology* **24**, 382001 (2013).
 27. Park, H. L., Lee, Y., Kim, N., Seo, D. G., Go, G. T. & Lee, T. W. Flexible Neuromorphic Electronics for Computing, Soft Robotics, and Neuroprosthetics. *Adv. Mater.* 1903558 (2019).

28. Rao, F., Ding, K. Y., Zhou, Y. X., Zheng, Y. H., Xia, M. J., Lv, S. L., Song, Z. T., Feng, S. L., Ronneberger, I., Mazzarello, R., Zhang, W. & Ma, E. Reducing the stochasticity of crystal nucleation to enable subnanosecond memory writing. *Science* **358**, 1423-1427 (2017).
29. Wang, C., Wu, H. Q., Gao, B., Wu, W., Dai, L. J., Li, X. Y. & Qian, H. Ultrafast RESET Analysis of HfO_x-Based RRAM by Sub-Nanosecond Pulses. *Adv. Electron. Mater.* **3**, 1700263 (2017).
30. Boyn, S., Chanthbouala, A., Girod, S., Carrétéro, C., Barthélémy, A., Bibes, M., Grollier, J., Fusil, S. & Garcia, V. Real-time switching dynamics of ferroelectric tunnel junctions under single-shot voltage pulses. *Appl. Phys. Lett.* **113**, 232902 (2018).
31. Havel, V., Fleck, K., Rösger, B., Rana, V., Menzel, S., Böttger, U. & Waser, R. Ultrafast switching in Ta₂O₅-based resistive memories. *Silicon Nanoelectronics Workshop SNW*, 82-83 (2016).

REVIEWERS' COMMENTS:

Reviewer #2 (Remarks to the Author):

The authors have revised the manuscript with large amount of new data. The quality of the paper has since been improved significantly. Some points that may be helpful to further polish the manuscript are listed as follows:

1. The protocol for measuring the endurance should be elaborated in detail. For example, in the current description, it is not clear after how many pulses was the resistance read during the measurements.
2. In the two layer perception simulation, the algorithm, namely SGD with BP would require that the weight updating is both linear and symmetric. Fig. S14 shows depression which is pretty linear, but how about potentiation? This part needs some more detailed description as well.
3. During inference, the authors states that a lower voltage gives higher accuracy because with a lower read voltage, the IV curves are more linear. Given that 10 mv range is fairly low (as compared with the 0.1 V the authors used for most of the reading), what is the signal to noise ratio going to be like?
4. There are some figures in the rebuttal letter, such as Figs, R7, 8, 9 deserve spots in SI. Fig. R.2 (Fig. S 5c) needs error bars for each data points.
5. u.c. (unit cell) is used quite often in the thin film growth community to measure the thickness. However, it is not a standard thickness unit and hence the corresponding thickness in SI should be given as well.
6. V_{bi} is NOT the notion for Schottky barrier height (should be capital- ϕ_B), it is commonly used for the built in potential. Please double check figures and equations in both the main text and SI, and make necessary corrections.

A list of changes

1. In **Line 221 of Page 12 of the Main Text in the merged PDF file**, the subsection title “Ultrafast ferroelectric based synapse: spike-timing-dependent plasticity (STDP)” was shortened to be “Ultrafast ferroelectric based synapse” in the revised manuscript.
2. In **Line 257 of Page 14 of the Main Text**, the subsection title “Band structure modulation”, was deleted.
3. In **Line 312 of Page 17**, the subsection title “High temperature endurance and retention properties” was deleted.
4. In **Line 333 of Page 18**, the subsection title “Artificial neural network simulation with FTJs” was deleted.
5. In **Lines 339-341 of Page 18**, the sentence “Accordingly, the effects of the cycle-to-cycle..... recognition accuracy were studied.” was added.
6. In **Supplementary Note 2 of the Supplementary information**, the sentence “Here, V_r is the reading voltage..... with V_r in between 0.01 V and 0.1 V.” was added.
7. In **Supplementary Note 5 of the Supplementary information**, the sentences “Here, for the first 100 cycles..... from 10^n to 10^{n+1} cycles.” were added.
8. In **Supplementary Note 5 of the Supplementary information**, the error bars for every data point in **Supplementary Fig. 5c** were added.
9. In **Supplementary Note 12 of the Supplementary information**, the conductance potentiation curve (**Supplementary Fig. 14**), the effects of the cycle-to-cycle variation (**Supplementary Fig. 15b**) and I - V nonlinearity (**Supplementary Fig. 16**) on simulated recognition accuracy were added.
10. In **the Main Text and the Supplementary information**, the previous notion for Schottky barrier height V_{bi} was replaced by Φ_B , and the thickness information of the BaTiO₃ \sim 2.4 nm was added.

There are also some minor revisions according to the editorial requests, which are not listed here.

Responses to Reviewer #2

Thank you very much for your pertinent comments and valuable suggestions on our manuscript. We have carefully revised the manuscript following your suggestions, and our responses to your comments are listed point by point as follows:

Comment 1. *The protocol for measuring the endurance should be elaborated in detail. For example, in the current description, it is not clear after how many pulses was the resistance read during the measurements.*

Answer 1: We thank the reviewer for pointing out this issue. To carry out the endurance measurements shown in Supplementary Fig. 5a, b, a function generator (Agilent 33220A) was utilized to generate square voltage pulses (± 3 V amplitude, 100 ns duration, 1 MHz repetition rate) to flip the ferroelectric polarization repeatedly, and the resistance (read at 0.1 V) was monitored after applying voltage pulses of defined numbers. For the first 100 cycles of measurements, the resistances were recorded for each cycle. Then, representative resistance switching measurements (for 10 cycles) were carried out every 10^n cycles ($n \geq 2$) from 10^n to 10^{n+1} cycles. For example, from 10^7 to 10^8 cycles ($n = 7$), resistance switching measurements (for 10 cycles) were carried out after 10^7 , 2×10^7 , 3×10^7 , 4×10^7 , 5×10^7 , 6×10^7 , 7×10^7 , 8×10^7 , and 9×10^7 cycles, respectively.

The corresponding discussions were added in **Supplementary Note 5** of the revised Supplementary Information, respectively.

Comment 2. *In the two layer perception simulation, the algorithm, namely SGD with BP would require that the weight updating is both linear and symmetric. Fig. S14 shows depression which is pretty linear, but how about potentiation? This part needs some more detailed description as well.*

Answer 2: We thank the reviewer for raising this important topic. According to your suggestion, the conductance evolutions for both depression and potentiation were measured, as shown in Fig. R1 in which the device shows 64 conductance states. It can be seen that the conduction potentiation is also linear. In addition, through a proper selection of the pulse sequences, the conductance potentiation (by applying voltage pulses from 0.61 V to 0.67 V with a step of 30 mV, 0.7 V to 1.28 V with a step of 20 mV, 1.3 V to 1.59 V with a step of 10 mV) can be very symmetric to the conductance depression (by applying voltage pulses with incremental amplitudes from -0.3 V to -1.8 V with a step of 30 mV, -1.9 V to -3.1 V with a step of 100 mV).

The corresponding discussions were added in **Supplementary Note 12** of the revised Supplementary Information, respectively.

Fig. R1 **a** Conductance vs. pulse number measured for 20 times. **b** Average conductance vs. pulse number. The error bar indicates the standard deviation of the cycle-to-cycle variations for each state.

Comment 3. *During inference, the authors states that a lower voltage gives higher accuracy because with a lower read voltage, the IV curves are more linear. Given that 10 mv range is fairly low (as compared with the 0.1 V the authors used for most of the reading), what is the signal to noise ratio going to be like?*

Answer 3: Thank you for raising this important topic. To evaluate the signal to noise ratios at different reading voltages, the electrical currents were tested by 10 times at different reading voltages for the ON and OFF states, as shown in Fig. R2a, b, respectively. The relative standard deviation (RSD) of current fluctuations can be calculated. It can be seen that a small RSD ($< 0.1\%$ for the ON state and $< 1\%$ for the OFF state) is observed at 10 mV, demonstrating the good signal to noise ratio of the FTJs. In addition, it is noted that the corresponding standard deviation relative to the entire conductance range Δ ($< 0.1\%$ for the ON state and $< 0.01\%$ for the OFF state) is much smaller than the experimental cycle-to-cycle variation ($\Delta \sim 2.0\%$, see Supplementary Fig. S14), and it thus has little effect on the simulated recognition accuracy (see Supplementary Fig. S15b).

The corresponding discussions were added in **Supplementary Note 2** of the revised Supplementary Information.

Fig. R2 Electrical currents read by 10 times at different reading voltages for **a** ON and **b** OFF states.

Comment 4. *There are some figures in the rebuttal letter, such as Figs, R7, 8, 9 deserve spots in SI. Fig. R.2 (Fig. S 5c) needs error bars for each data points.*

Answer 4: Thank you for your valuable suggestion. The previous Figs. R7, R8, and R9 were added as the **Supplementary Figs. 15b and 16**, and the corresponding discussions were added in **Supplementary Note 12** of the revised Supplementary Information. Error bars in **Supplementary Fig. S5c** were added for each data point.

Comment 5. *u.c. (unit cell) is used quite often in the thin film growth community to measure the thickness. However, it is not a standard thickness unit and hence the corresponding thickness in SI should be given as well.*

Answer 5: Thank you for your valuable suggestion. The thickness of the BaTiO_3 ~ 2.4 nm was added in the revised Main Text and Supplementary Information.

Comment 6. *V_{bi} is NOT the notion for Schottky barrier height (should be capital- ϕ_B), it is commonly used for the built in potential. Please double check figures and equations in both the main text and SI, and make necessary corrections.*

Answer 6: We appreciate the reviewer's careful review very much. According to your suggestion, the notion for Schottky barrier height was replaced by Φ_B in the revised Main Text and Supplementary Information.